# Genetic insights into the social organization of Neanderthals

Laurits Skov[1✉], Stéphane Peyrégne[1], Divyaratan Popli[1], Leonardo N. M. Iasi[1], Thibaut Devièse[2], Viviane Slon[1,3,4,5], Elena I. Zavala[1], Mateja Hajdinjak[1,6], Arev P. Sümer[1], Steffi Grote[1], Alba Bossoms Mesa[1], David López Herráez[1], Birgit Nickel[1], Sarah Nagel[1], Julia Richter[1], Elena Essel[1], Marie Gansauge[1], Anna Schmidt[1], Petra Korlević[1,7], Daniel Comeskey[8], Anatoly P. Derevianko[9], Aliona Kharevich[9], Sergey V. Markin[9], Sahra Talamo[10,11], Katerina Douka[12,13,14], Maciej T. Krajcarz[15], Richard G. Roberts[16,17], Thomas Higham[12,14], Bence Viola[18], Andrey I. Krivoshapkin[9], Kseniya A. Kolobova[9], Janet Kelso[1], Matthias Meyer[1], Svante Pääbo[1] & Benjamin M. Peter[1✉]

Genomic analyses of Neanderthals have previously provided insights into their population history and relationship to modern humans[1–8], but the social organization of Neanderthal communities remains poorly understood. Here we present genetic data for 13 Neanderthals from two Middle Palaeolithic sites in the Altai Mountains of southern Siberia: 11 from Chagyrskaya Cave[9,10] and 2 from Okladnikov Cave[11]—making this one of the largest genetic studies of a Neanderthal population to date. We used hybridization capture to obtain genome-wide nuclear data, as well as mitochondrial and Y-chromosome sequences. Some Chagyrskaya individuals were closely related, including a father–daughter pair and a pair of second-degree relatives, indicating that at least some of the individuals lived at the same time. Up to one-third of these individuals' genomes had long segments of homozygosity, suggesting that the Chagyrskaya Neanderthals were part of a small community. In addition, the Y-chromosome diversity is an order of magnitude lower than the mitochondrial diversity, a pattern that we found is best explained by female migration between communities. Thus, the genetic data presented here provide a detailed documentation of the social organization of an isolated Neanderthal community at the easternmost extent of their known range.

Neanderthals occupied western Eurasia from around 430,000 years ago[8,12] until their extinction around 40,000 years ago[13]. Genome-scale data have been reported for the skeletal remains of 18 individuals from 14 archaeological sites[1–8] spanning Neanderthal history across large parts of their known geographical range, which extends as far east as the Altai Mountains in southern Siberia. These data have yielded a broad overview of Neanderthal populations, indicating the existence of multiple distinct Neanderthal populations over time and space[1,2,14].

However, little is known about the genetic relationships and social organization within and between Neanderthal communities in any part of Eurasia during this time interval.

By 'social organization', we mean the size, sex composition and spatiotemporal cohesion of a community[15]. We define a community as a set of individuals that presumably lived together at the same location,

and reserve the term population for a broadly connected set of communities in a wider geographical area.

On the basis of fossilized footprints[16,17] and spatial patterns of site use[18], previous studies on the social organization of Neanderthal communities have suggested that Neanderthals probably lived in small communities. In addition, partial mitochondrial DNA (mtDNA) sequences from six adult Neanderthals have been used to suggest that Neanderthals may have been patrilocal[19], although this suggestion has been debated[20].

Here we explore the social organization of Neanderthals using nuclear, Y-chromosomal and mtDNA data from the remains of 13 individuals recovered from 2 sites located close to one another in southern Siberia (Russia)—Chagyrskaya and Okladnikov caves (Table 1 and Fig. 1a).

[1]Department of Evolutionary Genetics, Max Planck Institute for Evolutionary Anthropology, Leipzig, Germany. [2]European Centre for Research and Education in Environmental Geosciences (CEREGE), Aix-Marseille University, CNRS, IRD, INRAE, Collège de France, Aix-en-Provence, France. [3]Department of Anatomy and Anthropology Sackler, Faculty of Medicine, Tel Aviv University, Tel Aviv, Israel. [4]The Dan David Center for Human Evolution and Biohistory Research, Tel Aviv University, Tel Aviv, Israel. [5]Department of Human Molecular Genetics and Biochemistry, Sackler Faculty of Medicine, Tel Aviv University, Tel Aviv, Israel. [6]The Francis Crick Institute, London, UK. [7]Wellcome Sanger Institute, Hinxton, UK. [8]Oxford Radiocarbon Accelerator Unit, Research Laboratory for Archaeology and the History of Art, University of Oxford, Oxford, UK. [9]Institute of Archaeology and Ethnography, Russian Academy of Sciences, Novosibirsk, Russia. [10]Department of Chemistry G. Ciamician, Alma Mater Studiorum, University of Bologna, Bologna, Italy. [11]Department of Human Evolution, Max Planck Institute for Evolutionary Anthropology, Leipzig, Germany. [12]Department of Evolutionary Anthropology, Faculty of Life Sciences, University of Vienna, Vienna, Austria. [13]Department of Archaeology, Max Planck Institute for the Science of Human History, Jena, Germany. [14]Human Evolution and Archaeological Sciences Forschungsverbund, University of Vienna, Vienna, Austria. [15]Institute of Geological Sciences, Polish Academy of Sciences, Warsaw, Poland. [16]Centre for Archaeological Science, School of Earth, Atmospheric and Life Sciences, University of Wollongong, Wollongong, New South Wales, Australia. [17]Australian Research Council (ARC) Centre of Excellence for Australian Biodiversity and Heritage, University of Wollongong, Wollongong, New South Wales, Australia. [18]Department of Anthropology, University of Toronto, Toronto, Ontario, Canada. ✉e-mail: laurits_skov@eva.mpg.de; benjamin_peter@eva.mpg.de

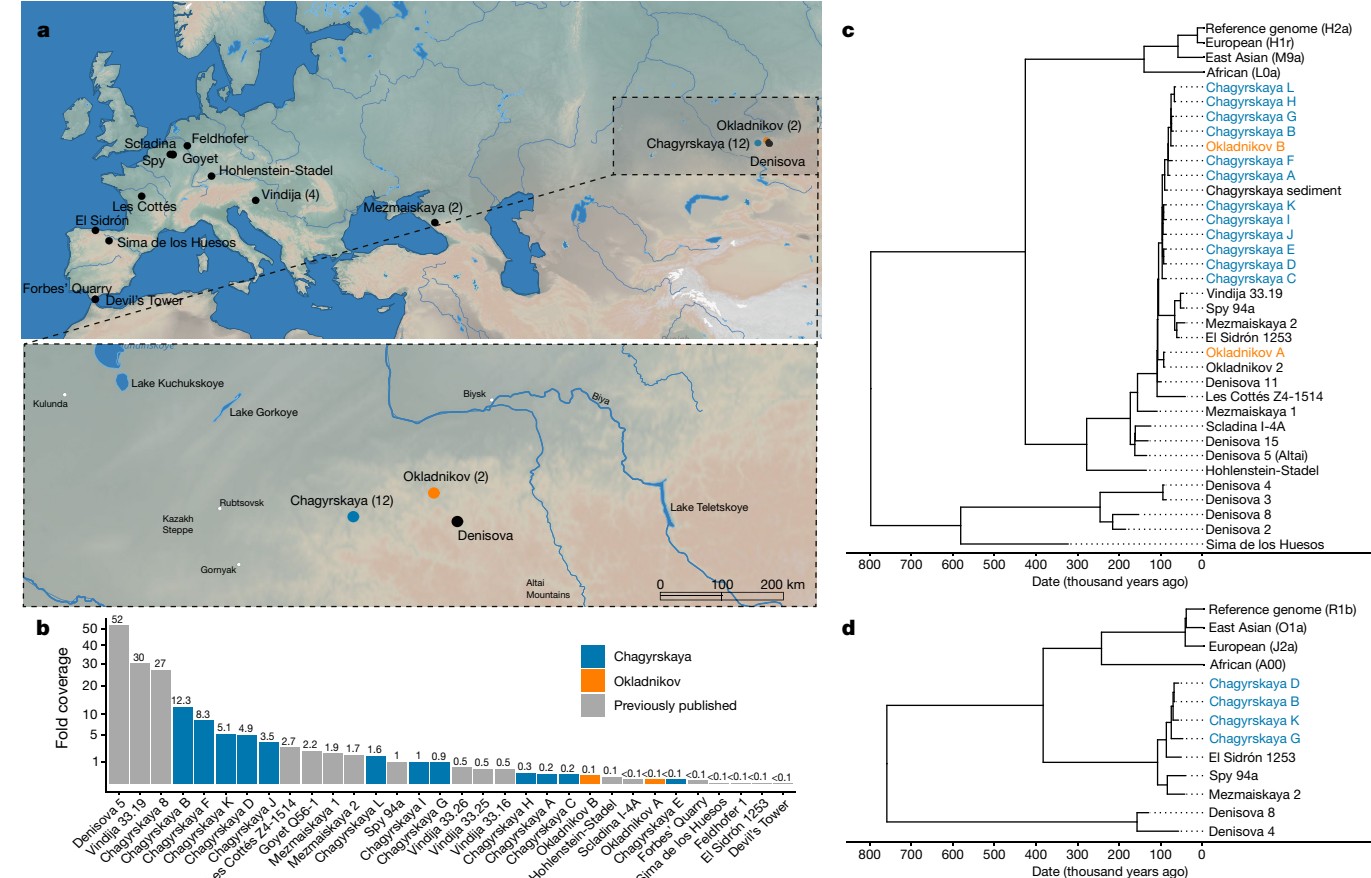

**Fig. 1 | Neanderthal sites and genomic information. a**, Locations of all of the sites with Neanderthal remains (the number of individuals is given in parentheses for sites with multiple individuals) from whom nuclear DNA has been extracted, with a close-up of the Chagyrskaya and Okladnikov caves in the Altai region of southern Siberia. **b**, Nuclear genomes ranked by the extent of coverage and colour-coded by site (blue, Chagyrskaya from this study; orange, Okladnikov from this study; grey, published previously in refs. [1–8]). **c**, Maximum-likelihood tree for mtDNA sequences from the Neanderthal individuals included in this study in the context of known hominin variation. The reference genome is rCRS and the accession numbers for the present-day humans are East Asian (AF346973), European (AF346981) and African (AF381988). Okladnikov 2 refers to the mtDNA sequence in ref. [41] (this specimen is listed as Okladnikov 14 in Extended Data Table 1). Data from refs. [1–4,6,30,41,42–49]. **d**, Maximum-likelihood tree based on consensus calling of 6.9 Mb of the Y chromosome of four Chagyrskaya individuals with coverage of more than onefold, along with previously published Y-chromosome data from three Neanderthals, two Denisovans and four present-day humans. The reference genome is hg19. Data from refs. [26,50–53]. In **c** and **d**, the haplogroups are shown for present-day human populations.

## Archaeological sites and remains

The Chagyrskaya and Okladnikov caves, located in the foothills of the Altai Mountains (Fig. 1a and Extended Data Figs. 1 and 2), are thought to have been used mainly as short-term hunting camps[11,21]. They are two of three known sites at which a distinctive Sibiryachikha Middle Palaeolithic industry has been found (the third being Upper Sibirya-chikha Cave)[9,10,22,23] (Supplementary Fig. 1.6). The Sibiryachikha industry at Chagyrskaya and Okladnikov caves is distinct from the Middle Palaeolithic industry at Denisova Cave (located around 100 km to the east), where Neanderthal remains have also been found[2].

The Neanderthal occupation deposits at Chagyrskaya Cave accumulated between 59,000 and 51,000 years ago, as indicated by optical dating of sediments and radiocarbon dating of bison bones[10]. We obtained additional radiocarbon ages from two pieces of charcoal and a Neanderthal bone (Chagyrskaya 9), all of which were older than 50,000 years before present (Supplementary Table 1.3). These ages are compatible with a short period of deposition (a few millennia or less), which is consistent with the presence of similar archaeological industry in all Neanderthal layers[10] (Extended Data Fig. 2).

For Okladnikov Cave, we constrained the timing of Neanderthal occupation using hydroxyproline-based single amino-acid radiocarbon ages

for three Neanderthal specimens (including Okladnikov 15) (Table 1 and Extended Data Table 1), which indicated that they were at least 44,000 years old (Supplementary Table 1.4). Our age estimates are consistent with uranium-series ages for animal bones and support previous suggestions that younger radiocarbon ages obtained from the collagen fraction reflect an incomplete removal of contaminants[24] (Supplementary Information section 1). Therefore, the archaeological and chronological data suggest that the Neanderthals that occupied these two sites may have been broadly contemporaneous.

Previous analyses of high-coverage genomes of a Neanderthal from Chagyrskaya Cave (Chagyrskaya 8) and an earlier Neanderthal from Denisova Cave (Denisova 5, the 'Altai Neanderthal') revealed that they belonged to different populations[5]. A first-generation offspring (Denisova 11) of a Neanderthal mother and a Denisovan father revealed that the Neanderthal mother was more similar to Chagyrskaya 8 than she was to other Neanderthals[5,25].

## Data acquisition and sex determination

We sampled 1–64 mg of tooth or bone powder from 17 specimens from Chagyrskaya Cave and 10 specimens from Okladnikov Cave. Of these, 15 from Chagyrskaya and 2 from Okladnikov yielded ancient DNA

**Table 1 | Neanderthals from Chagyrskaya and Okladnikov Caves included in this study**

| Individual | Bone/tooth ID | Age | Anatomical element | Genetic sex | Relationship to other individual(s) |
|---|---|---|---|---|---|
| Chagyrskaya A | Chagyrskaya 1 | 8–12 (D) | Deciduous lower left canine | Male | Second-degree relation of Chagyrskaya L |
| Chagyrskaya B | Chagyrskaya 2 | 3–5 | Atlas (first cervical vertebra) | Male | |
| Chagyrskaya C | Chagyrskaya 6 | Adult | Right mandible fragment with canine to $M_2$ | Male | |
| Chagyrskaya C | Chagyrskaya 14 | Adult | Lower left second incisor | Male | |
| Chagyrskaya D | Chagyrskaya 7 | Adult? | Thoracic vertebral process fragment | Male | Father of Chagyrskaya H; possible first-degree relation of/identical to Chagyrskaya E |
| Chagyrskaya E? | Chagyrskaya 9 | Adult | Left proximal ulna fragment | Male | Possible first-degree relation of/identical to Chagyrskaya D |
| Chagyrskaya F | Chagyrskaya 12 | Adult | Left third premolar | Female | |
| Chagyrskaya F | Chagyrskaya 8[a] | Adult | Distal phalanx of the hand (high-coverage genome) | Female | |
| Chagyrskaya G | Chagyrskaya 13 | 10–15 | Left upper first incisor | Male | |
| Chagyrskaya G | Chagyrskaya 19 | 9–11 (D) | Deciduous left upper second molar | Male | |
| Chagyrskaya G | Chagyrskaya 63 | 9–14 | Upper left second molar crown | Male | |
| Chagyrskaya H | Chagyrskaya 17 | 15–20? | Right lower fourth premolar | Female | Daughter of Chagyrskaya D |
| Chagyrskaya I | Chagyrskaya 18 | 9–11 (D) | Deciduous left upper $M^1$ | Female | |
| Chagyrskaya J | Chagyrskaya 20 | 7–12 (D) | Deciduous right upper canine | Female | |
| Chagyrskaya K | Chagyrskaya 41 | Adult | Right lower third premolar | Male | |
| Chagyrskaya L | Chagyrskaya 60 | Adult | Middle phalanx of the hand | Female | Second-degree relation of Chagyrskaya A |
| Okladnikov A | Okladnikov 11 | 7–11 | Proximal half of a juvenile femur | Male | |
| Okladnikov B | Okladnikov 15 | Adult | Right distal humerus fragment | Female | |

Ages represent age-at-death estimates based on anatomical features, with the exception of the deciduous teeth (D); for these naturally exfoliated teeth, age is the time of tooth loss. Details are provided in Supplementary Information section 1.

[a]A high-coverage genome for Chagyrskaya 8 has been published previously[5].

(Table 1, Extended Data Table 1 and Supplementary Data 1), from which we generated a total of 85 single-stranded DNA libraries (Supplementary Information section 2). All of the libraries were enriched for mtDNA sequences (Supplementary Information section 3) and 49 libraries (selected for high sequence yields and low levels of present-day human contamination) were enriched for nuclear DNA using a newly designed nuclear-capture array containing 643,472 transversion polymorphisms across the genome (Supplementary Information section 5). In the array, 271,306 sites vary among the 4 published high-coverage archaic individuals (three Neanderthals and one Denisovan)[2,3,5,14] and 372,166 sites segregate in present-day African populations or are fixed between present-day humans and archaic hominins. The average nuclear DNA coverage for each fossil ranges from 0.04- to 12.3-fold (Fig. 1b), and present-day human contamination estimates range from 0.1% to 3.2% (Supplementary Table 5.4).

We determined the genetic sex of the 17 remains using the difference in coverage between the X chromosome and autosomes (Supplementary Fig. 5.5) and found that 6 remains stemmed from females. For the 11 male remains, we enriched the libraries for around 6.9 megabases (Mb) of Y-chromosome sequence[26] (Supplementary Information section 4), yielding coverages ranging between 0.02- and 42.2-fold (Supplementary Table 4.3).

## Identification of relatives

To determine whether any of the remains originated from related individuals, we computed the nuclear DNA divergence between the 17 remains by randomly sampling 1 allele from 250,785 sites in the capture array that were variable in the high-coverage archaic individuals (excluding variants specific to Chagyrskaya 8) (Supplementary Information section 5). The divergence will be lower for related individuals

because they have inherited parts of their genomes from the ancestors they share in the recent past. We normalized this divergence ($p_0$) by a median DNA divergence among all comparisons. Using this approach[27], we can detect up to second-degree relationships; we consider everything beyond that as unrelated. We expect $p_0 = 1$ for remains who are more distantly related than second-degree relatives, $p_0 = 0.875$ for second-degree relatives, $p_0 = 0.75$ for first-degree relatives and $p_0 = 0.5$ for remains from monozygotic twins or the same individual[27]. We also investigated mtDNA heteroplasmies (that is, when mitochondria carried by an individual differ in their DNA sequence) (Supplementary Table 3.2) to identify close genetic relationships[28]. As heteroplasmies can be transmitted from mother to child and typically persist for less than three generations[29], their presence in different remains would indicate that they come from the same or maternally closely related individuals. To differentiate between remains (that is, between skeletal and dental samples) and individuals, we denote the former with numbers and the latter with letters (Table 1).

We found a deciduous tooth (Chagyrskaya 19) and two permanent teeth (Chagyrskaya 13 and Chagyrskaya 63). Surprisingly, despite their different developmental stages, the genetic data suggest that they belonged to the same individual (Chagyrskaya G; average $p_0 = 0.53$) (Extended Data Fig. 3a). In agreement with this, all three teeth stemmed from a male and carried identical mtDNAs, including a heteroplasmy at position 3,961 at similar frequencies of 60.7–78.5% (Supplementary Table 3.2). The almost completely resorbed root of the deciduous tooth suggests that it was naturally exfoliated (Supplementary Information section 1). On the basis of patterns of wear and root development, we inferred that the permanent teeth came from a 9–15-year-old individual and that this male probably died around the time the deciduous tooth was lost.

We also identified two further sets of individuals with multiple fossils: Chagyrskaya C is represented by both Chagyrskaya 6, a mandible,

and Chagyrskaya 14, a permanent incisor (Supplementary Information section 1), as evidenced by the morphological fit, identical mtDNA sequences (including a shared heteroplasmy) and low nuclear divergence ($p_0 = 0.65$; 95% confidence interval, 0.34–0.78) (Fig. 1c, Extended Data Fig. 3a and Supplementary Tables 3.2 and 7.1). Similarly, Chagyrskaya F is represented by both Chagyrskaya 12 and the previously sequenced[5] Chagyrskaya 8 ($p_0 = 0.46$; 95% confidence interval, 0.41–0.46) (Supplementary Table 7.1).

One adult male individual, Chagyrskaya D, was closely related to multiple other individuals in the group. We found a first-degree relationship between him and Chagyrskaya H, who is an adolescent female ($p_0 = 0.77$; 95% confidence interval, 0.72–0.82). There are three possible male–female combinations for first-degree relatives: mother–son, brother–sister or father–daughter. However, since the two individuals carry different mitochondrial genomes (Fig. 1c), we concluded that Chagyrskaya H was the daughter of Chagyrskaya D.

In addition, his mtDNA was identical to that of two other males, Chagyrskaya C and Chagyrskaya E (Supplementary Table 3.2), including a shared mtDNA heteroplasmy at position 545 (G>A) with a frequency of A of 42–54% for Chagyrskaya D, 20–41% for Chagyrskaya E and 23–30% for Chagyrskaya C. Therefore, these individuals were probably close maternal relatives (for example, they could have shared a grandmother and thus might have been fourth-degree relatives). However, the extent of the relationship between Chagyrskaya C and Chagyrskaya D is beyond the resolution of our approach ($p_0 = 1.05$; 95% confidence interval, 0.94–1.16). Chagyrskaya E has low coverage (Supplementary Table 5.4) and high amounts of human and nonhuman contamination (Supplementary Table 5.3). After correcting for nonhuman contamination (Supplementary Table 7.1), we identified Chagyrskaya E as either a first-degree relative of or identical to Chagyrskaya D ($p_0 = 0.64$; 95% confidence interval, 0.48–0.79). As we cannot be confident that Chagyrskaya E is a distinct individual, we removed the sample from further analysis.

The close relationships among Chagyrskaya C, D and H imply that they were contemporaneous. In addition, we found that Chagyrskaya A (male) and L (female) are second-degree relatives ($p_0 = 0.85$; 95% confidence interval, 0.77–0.91). Although the sparse data prevented us from determining the exact relationship, they must also have lived close in time (Extended Data Fig. 3b). The genetic divergence between each group of contemporaneous individuals and the other six Chagyrskaya individuals were not significantly different from each other (Wilcoxon rank-sum test, both $P > 0.26$) (Supplementary Table 7.4). In addition, the contemporaneous father–daughter pair carried the highest number of differences among all mtDNA sequences, implying that there was no substantial temporal structure in the mtDNA diversity. Taken together, the data supported the hypothesis that all eleven Chagyrskaya Neanderthals were part of the same community.

The two Okladnikov remains were unrelated to each other ($p_0 = 1.14$; 95% confidence interval, 0.90–1.38) and also not related to any individual from Chagyrskaya Cave. In fact, the pairwise genetic divergence among the Chagyrskaya individuals was lower ($p_0 = 1.0$; 95% confidence interval, 0.99–1.02) than that between individuals from Chagyrskaya and Okladnikov caves ($p_0 = 1.06$; Wilcoxon rank-sum test, $P = 8.6 \times 10^{-5}$) (Extended Data Fig. 3a and Supplementary Table 7.3). This indicates that the Okladnikov Neanderthals were not part of the Chagyrskaya Neanderthal community represented by the 11 individuals for which we obtained DNA. However, the mtDNA of Okladnikov B is identical to that of Chagyrskaya G (Fig. 1c). Because mutations accumulate over time, identical mtDNA between individuals implies that these two individuals lived within a few thousand years of each other (Supplementary Table 3.9). In addition, among the previously published sediment mtDNA samples from Chagyrskaya Cave, 2 of the 38 samples were more similar to Okladnikov A than they were to any Chagyrskaya Neanderthal[30]. This suggests there was some connection between the communities occupying the two caves.

## Relationships to other populations

To explore how the Chagyrskaya and Okladnikov individuals are related to other Neanderthals, we investigated the extent to which they share nucleotide variants with the previously published high-quality Neanderthal genomes. All 13 newly sequenced individuals shared most variants with the high-coverage genome from Chagyrskaya Cave (Chagyrskaya 8)[5] and were more similar to the around 50,000-year-old Neanderthal genome from Vindija Cave (Vindija 33.19)[3] in Croatia than to the 91,000–130,000-year-old Altai Neanderthal (Denisova 5) from Denisova Cave[2] (Extended Data Fig. 4). Therefore, although the communities from Chagyrskaya and Okladnikov caves were genetically distinct, they all appear equally related to European Neanderthals and were part of the same Neanderthal population; no individual showed evidence of recent gene flow from other Neanderthal populations.

We identified 5,416 variants in the 6.9 Mb sequence of the Y chromosome that varied among the Y chromosomes of the seven male individuals, three Neanderthals, two Denisovans and four present-day humans (Supplementary Table 4.7). For three individuals, we obtained only low-coverage sequences (0.03- to 0.3-fold), whereas the other four individuals yielded higher coverages (1.75- to 42.2-fold) (Supplementary Table 4.3).

We constructed a phylogenetic tree that incorporated the four higher-coverage Y-chromosome sequences from Chagyrskaya Cave, along with those of three other Neanderthals, two Denisovans and four present-day humans (Fig. 1d and Supplementary Table 4.7). Among Neanderthals, all four Chagyrskaya sequences form a clade, but they are more similar to El Sidrón 1253 (Spain) than to the geographically closer Mezmaiskaya 2 (northern Caucasus, Russia) (Fig. 1d). This absence of geographical structure is consistent with a fairly rapid expansion of Neanderthals around 100,000–115,000 years ago[30]. Both the late European Neanderthals and the Chagyrskaya and Okladnikov Neanderthals are descendants of this population.

The number of recovered Y-chromosome sequences from the remaining three individuals were not sufficient for constructing a phylogenetic tree, but at positions at which the Neanderthal Y chromosomes differed from each other, all three sequences shared more derived variants with the other Chagyrskaya Y chromosomes than with other Neanderthal Y chromosomes (Supplementary Table 4.9).

On the basis of the differences in coverage in windows of 10 kilobases (kb), we detected 3 deletions and 5 duplications (20–2,000 kb and 10–200 kb in size, respectively) (Supplementary Table 4.4) on the Neanderthal Y chromosomes. The largest deletion was found in Mezmaiskaya 2 and spans the AMELY-encoding gene. Because proteomic approaches use the presence of AMELY peptides to determine whether a bone stems from a male individual[31], males who carry this deletion would be misclassified as females using this approach (Extended Data Fig. 5).

The mtDNA and Y chromosomes track only single loci, so autosomal genetic analyses are necessary to investigate details of gene flow. Gene flow between Neanderthals and Denisovans in the Altai Mountains has been observed in the nuclear genome of an individual (Denisova 11) who lived 79,000–118,000 years ago and had a Neanderthal mother and a Denisovan father[32]. It has also been estimated that the amount of Denisovan ancestry in Chagyrskaya 8 is around 0.09% and that the admixture event occurred 24,300 ± 14,100 years before Chagyrskaya 8 lived[33]. To investigate whether the timing of admixture is consistent across the other Chagyrskaya individuals, we looked for portions of their genomes that are more similar to the Denisovan genome than to the Altai or Vindija Neanderthals[33]. With this analysis, we identified 11 segments of Denisovan ancestry across 5 Chagyrskaya individuals that are longer than 0.2 centimorgans (cM) (Supplementary Table 6.2). These segments span 3.2 cM (2.7 Mb), with the longest at 1.5 cM (746 kb) found in Chagyrskaya A (Supplementary Fig. 6.2). On the basis of the lengths of these segments, we estimate that the admixture event

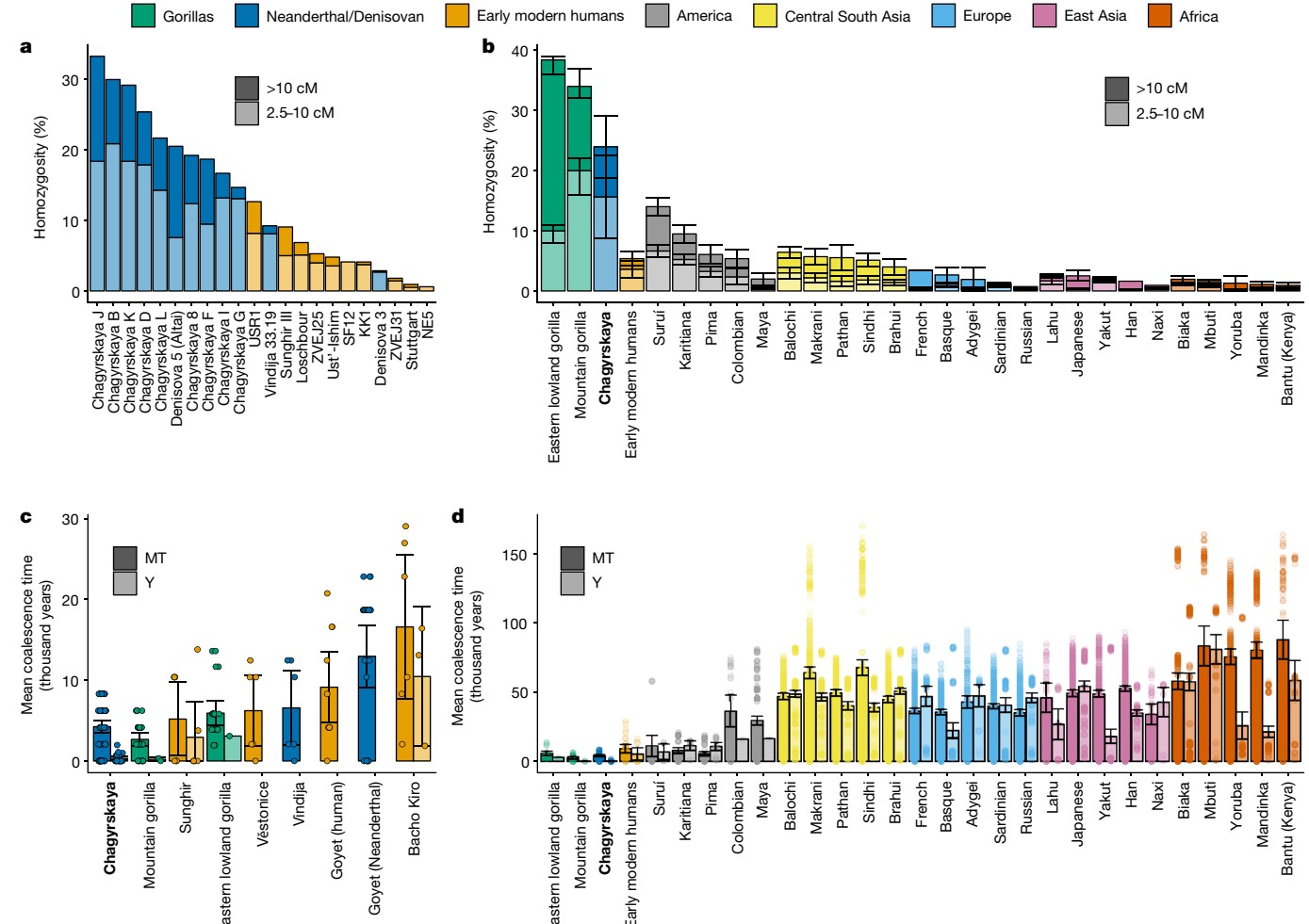

**Fig. 2 | Genomic diversity for Chagyrskaya Neanderthals compared with other hominids.** Neanderthal (blue), early modern human (orange) and present-day gorilla (green) populations are coloured the same throughout the figure. Present-day human populations are coloured according to the geographical region (see colour key). **a**, The proportion of the genome that is in homozygous tracts longer than 10 cM (dark) and tracts between 2.5 and 10 cM (light colour) for ancient individuals (early modern humans, Neanderthals and Denisovans). **b**, Average proportion of the genome that is homozygous for Chagyrskaya Neanderthals, early modern humans (grouped together) and present-day human and gorilla populations[37]. Data are mean ± 95% confidence intervals for the estimates of the mean. The sample size is equal to that of the mtDNA sequences listed below. **c**, Mean coalescence time for mtDNA (MT) and Y chromosome (left and right bars of each pair, respectively) for Neanderthal, early modern human and gorilla populations. **d**, Mean coalescence time for early modern humans (grouped together) and present-day human and gorilla populations. **c**,**d**, Data are mean ± 95% confidence intervals and points are all

pairwise comparisons. The number of Y chromosome and mtDNA-genomes used in pairwise comparisons for each population is as follows: Neanderthal and Denisovan, Chagyrskaya (MT = 12, Y = 6), Vindija (MT = 4, Y = 0), Goyet (Neanderthal) (MT = 7, Y = 0); early modern humans, Sunghir (MT = 4, Y = 4), Věstonice (MT = 4, Y = 0), Goyet (MT = 5, Y = 0), Bacho Kiro (MT = 4, Y = 3), which combined is (MT = 17, Y = 7); gorillas, mountain gorilla (MT = 8, Y = 3), eastern lowland gorilla (MT = 7, Y = 2); Americas, Suruí (MT = 9, Y = 4), Karitiana (MT = 13, Y = 5), Pima (MT = 14, Y = 7), Colombian (MT = 8, Y = 2), Mayan (MT = 22, Y = 2); central South Asia, Balochi (MT = 25, Y = 24), Makrani (MT = 26, Y = 20), Pathan (MT = 25, Y = 19), Sindhi (MT = 25, Y = 20), Brahui (MT = 26, Y = 25); Europe, French (MT = 29, Y = 11), Basque (MT = 24, Y = 15), Adygei (MT = 17, Y = 7), Sardinian (MT = 29, Y = 15), Russian (MT = 26, Y = 16); East Asia, Lahu (MT = 9, Y = 7), Japanese (MT = 28, Y = 19), Yakut (MT = 26, Y = 18), Han (MT = 34, Y = 15), Naxi (MT = 9, Y = 6); Africa, Biaka (MT = 23, Y = 22), Mbuti (MT = 14, Y = 10), Yoruba (MT = 23, Y = 11), Mandinka (MT = 23, Y = 14), Bantu (Kenya) (MT = 12, Y = 10).

happened 30,000 ± 18,000 years before the Chagyrskaya individuals lived, which is consistent with the previous estimate (Supplementary Fig. 6.3).

Denisova Cave was occupied by both Neanderthals and Denisovans around the same time that Neanderthals inhabited Chagyrskaya Cave[34,35]. However, the stone artefact industry at Denisova Cave lacks the characteristics of the Sibiryachikha variant found at Chagyrskaya Cave[10]. Accordingly, despite the proximity of the two caves and the presence of an offspring of a Neanderthal mother and a Denisovan father in Denisova Cave some tens of millennia before Chagyrskaya Cave was occupied[25], we find no evidence of gene flow from Denisovans to the Chagyrskaya Neanderthals in the last 20,000 years before the Chagyrskaya individuals lived (Supplementary Information section 6).

## Inferring social organization

We investigated the community and population size of the Chagyrskaya Neanderthals through time using genomic segments of homozygosity from 8 individuals (those with more than 0.9-fold genomic coverage) (Supplementary Information section 9). Long segments of homozygosity (greater than 10 cM) in an individual imply that their parents shared a very recent common ancestor around ten generations ago and were, therefore, probably part of a small community[5,36]. In addition, the overall proportion of the genome with intermediate length segments of homozygosity (2.5–10 cM) is informative of the size of the population over a slightly longer time frame (around 10–40 generations).

Previous analyses of high-coverage Neanderthal genomes from the Altai mountains revealed that around 16.7% of the genome of Denisova 5 (ref. [2]) and 19.3% of the genome of Chagyrskaya 8 (ref. [5]) had intermediate and long segments of homozygosity. One explanation for these patterns is that their parents were second-degree relatives[2] against a background of unrelated individuals, in which case we would expect most other individuals to have fewer homozygous segments. Alternatively, these data could be due to small local communities[5], in which case all individuals, except recent immigrants and their descendants, would have extensive segments of homozygosity.

In all 8 individuals with sufficient coverage, we observed that 1.6–14.9% of the genome had long segments of homozygosity and 9.5–20.5% had intermediate segments of homozygosity (Fig. 2a and Supplementary Table 9.2). We note that both proportions were probably underestimates owing to difficulties in identifying runs of homozygosity at lower coverages (Supplementary Table 9.1). Because we find high amounts of homozygosity in all individuals, we conclude that the local community size of the Chagyrskaya Neanderthals was small. The amount of homozygosity is also similar to the amount found in the genomes of present-day mountain gorillas[37] (Fig. 2b), an endangered species that lives in small communities of 4–20 individuals[38], in which it has been observed that matings between second-degree-related individuals are rare[39].

To further investigate the social organization of the Chagyrskaya Neanderthals, we contrasted the diversity of the 11 maternally inherited mtDNA sequences with the 6 paternally inherited Y-chromosome sequences. In a randomly mating population without sex-biased processes, the average coalescence time is expected to be the same for both uniparental markers. However, the observed average coalescent time for the Y chromosome (446 years; 95% confidence interval, 113–1,116 years) is significantly lower than that of the mitochondrial genome (4,348 years; 95% confidence interval, 2,043–6,196 years; Wilcoxon rank-sum test, $P = 4.1 \times 10^{-5}$). In a comparison with 47 modern human populations and 10 great ape subspecies, the Chagyrskaya Neanderthals have among the lowest ratios of Y-chromosome-to-mtDNA coalescence time, with only mountain gorillas having a more extreme ratio (Extended Data Fig. 6). We caution that similar ratios between apes and Neanderthals do not necessarily mean that the communities have the same social organization, as there are multiple caveats. First, the great ape data are very heterogeneous—for example, although some great apes were born in the wild, others were born in captivity (that is, in artificial communities) and often the sample sizes were very small (Supplementary Table 8.1). Second, several different scenarios may lead to similar Y-chromosome-to-mtDNA ratios. These include: differences in male and female generation times, a skewed offspring distribution among males (that is, a subset of males father the majority of the children) and female-biased migration. To test the relative importance of these processes, we simulated a large number of combinations of these scenarios, fitting the diversity of Y chromosomes and mtDNA and their ratio to the observed data (Supplementary Information section 8). We approximated the likelihood of each scenario using simulations as the proportion of simulated datasets that are within the 95% confidence intervals of the observed data. We then used the Akaike information criterion (AIC) to rank different scenarios (Supplementary Table 8.5).

The best-fitting scenarios (AIC = 6.2) assumed a community size of 20 individuals, with 60–100% of the females being migrants from another community (Supplementary Table 8.4). However, the shared heteroplasmy between Chagyrskaya C and Chagyrskaya D suggests that at least some females remained with the group they were born in. Scenarios that include only skewed offspring distributions explain the data less well (AIC = 7.4) and require large community sizes of 300 individuals. Scenarios with both skewed offspring distributions and female migrations does not improve the fit (AIC = 8.5) obtained by assuming migration-bias alone. Scenarios that include only differences in generation time fit the data poorly (AIC = 8.5) and require parameter settings that seem unrealistic (for example, females would need to be on average twice as old as males, Supplementary Table 8.4). Previous estimates of Neanderthal community sizes range from 3 to 60 individuals[5,16,17,19] and, in this range, the best fitting scenarios include female migration (Supplementary Fig. 8.4). This result suggests that female-biased migration was a major factor in the social organization of the Chagyrskaya Neanderthal community.

## Conclusion

We present genetic data from 13 Neanderthals, making this one of the largest genetic studies of a Neanderthal population. For the first time, to our knowledge, we document familial relationships between Neanderthals, including a father-and-daughter pair.

The high degree of homozygosity in all individuals is similar to what is seen in mountain gorillas[40], consistent with Neanderthals in the Altai living in small communities. Furthermore, based on the shorter average coalescent time for the Y chromosomes than for the mtDNA and shared mtDNA variants between Chagyrskaya and Okladnikov individuals, we suggest that these small Neanderthal communities were predominantly linked by female migration.

Our findings raise questions as to whether the characteristics of the Altai communities are related to their isolated geographical location at the easternmost extremity of the known range of Neanderthals (especially because the population size at Vindija Cave was probably larger[5]), or whether they are characteristic of Neanderthal communities more broadly.

Future studies should, therefore, when possible, aim to sample multiple individuals from additional Neanderthal communities in other parts of Eurasia to shed further light on the social organization of our closest evolutionary relatives.

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

## Methods

No statistical methods were used to predetermine sample size. The experiments were not randomized and the investigators were not blinded to allocation during experiments and outcome assessment. A detailed description of all analyses carried out in this study is included in the Supplementary Information. Permission to work on the archaeological specimens was granted based on a written agreement of scientific cooperation signed in 2018 by the Federal State Budgetary Institution of Science–Institute of Archaeology and Ethnography, Siberian Branch of the Russian Academy of Sciences and the Max Planck Institute for Evolutionary Anthropology.

### Reporting summary

Further information on research design is available in the Nature Research Reporting Summary linked to this article.

### Data availability

Raw data for each library are available in the European Nucleotide Archive under accession number PRJEB55327. Mapped BAM files for all specimens and individuals, VCF files, consensus FASTA mtDNA sequences and a multiple alignment of all mtDNA can be downloaded from http://ftp.eva.mpg.de/neandertal/ChagyrskayaOkladnikov/.

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

**Acknowledgements** We thank D. Lucas and C. Logan for comments on an earlier version of the manuscript; H. Temming for performing computed tomography scans; S. Sawyer and J. Krause, who drilled the bones of Chagyrskaya 2 and Chagyrskaya 6; R. Schultz for helping to recover the digital photographs of the samples and uploading the primary data files. V.S. was funded by the Alon Fellowship, M.H. was funded by Marie Skłodowska Curie Action (MSCA-IF-EF-ST LIF, "ORIGIN" no. 844014), S.T. was funded by the European Research Council Horizon 2020 Research and Innovation Programme grant "RESOLUTION" (no. 803147), K.D. was funded by the European Research Council Horizon 2020 Research and Innovation Programme grant "FINDER" (no. 715069), R.G.R. was funded by the Australian Research Council fellowship FL130100116, T.H. was funded by the European Research Council Seventh Framework Programme (FP7/2007-2013) grant 324139 "PalaeoChron", K.A.K. was funded by Russian Science Foundation, project N 21-18-00376, M.T.K. was funded by National Science Centre, Poland, grant 2018/29/B/ST10/00906, B.V. was funded by Social Sciences and Humanities Research Council, Canada, Insight grant 435-2018-0943. This project was funded by the European Research Council (grant agreement no. 694707 to S. Pääbo).

**Author contributions** L.S., S. Pääbo and B.M.P. designed the study. A.P.D., A.K., S.V.M., A.I.K. and K.A.K. collected samples. T.D., V.S., M.H., B.N., S.N., J.R., E.E., M.G., A.S., P.K., D.C., S.T., T.H. and B.V. performed laboratory experiments and/or analysis. L.S., S. Peyrégne, D.P., L.N.M.I., T.D., V.S., E.I.Z., M.H., A.P.S., S.G., A.B.M., D.H.L., D.C., A.P.D., A.K., S.V.M., S.T., K.D., M.T.K., R.G.R., T.H., B.V., A.I.K., K.A.K. and B.M.P. performed analyses. L.S., S. Peyrégne, D.P., L.N.M.I., T.D., V.S., E.I.Z., M.H., A.P.S., A.B.M., D.H.L., S.T., K.D., M.T.K., R.G.R., T.H., B.V., A.I.K., K.A.K., J.K., M.M., S. Pääbo and B.M.P. wrote the manuscript with input from all authors. A.P.D., A.K., S.V.M., K.D., M.T.K., R.G.R., T.H., B.V., A.I.K. and K.A.K. provided archaeological, stratigraphical and geochronological context and interpretation.

**Funding** Open access funding provided by Max Planck Society.

**Competing interests** The authors declare no competing interests.

**Additional information**
**Correspondence and requests for materials** should be addressed to Laurits Skov or Benjamin M. Peter.

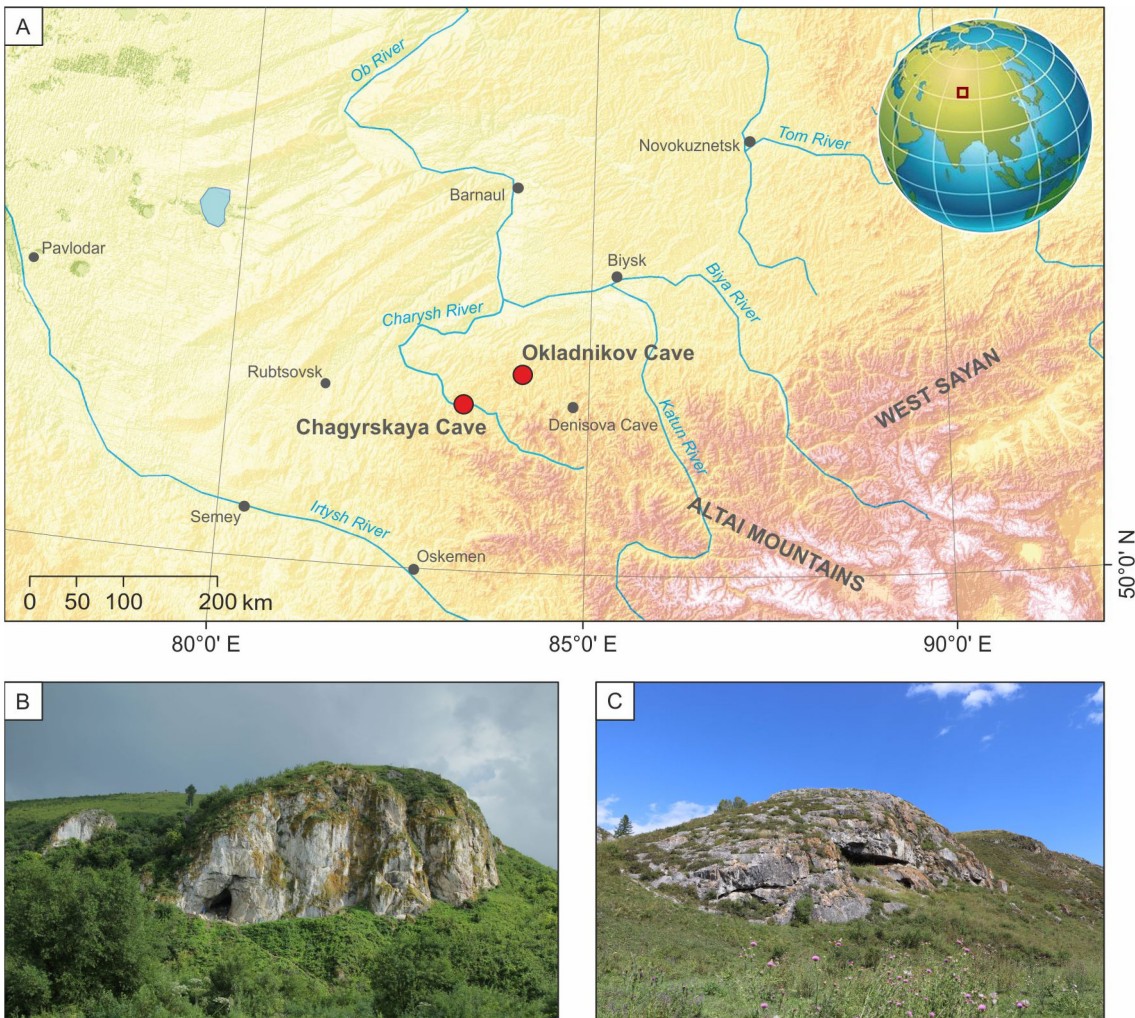

**Extended Data Fig. 1 | Chagyrskaya and Okladnikov Caves. A**, Location map of Chagyrskaya and Okladnikov Caves in the Altai region of southern Siberia. Views of the **B**, north-facing entrance to Chagyrskaya Cave and **C**, south-facing entrance to Okladnikov Cave.

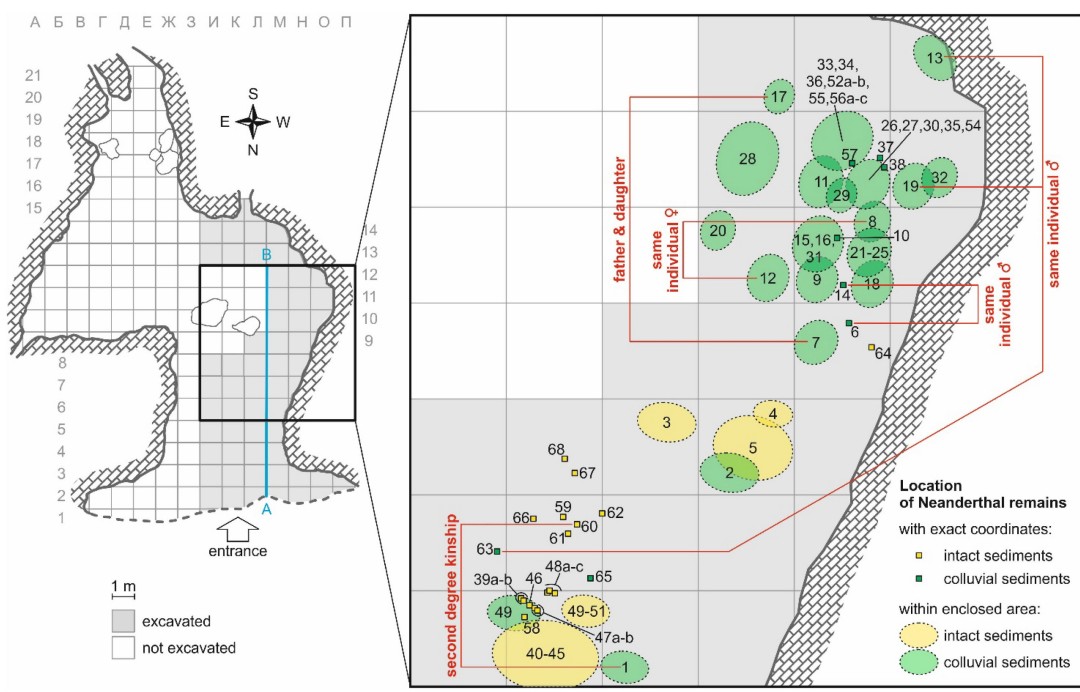

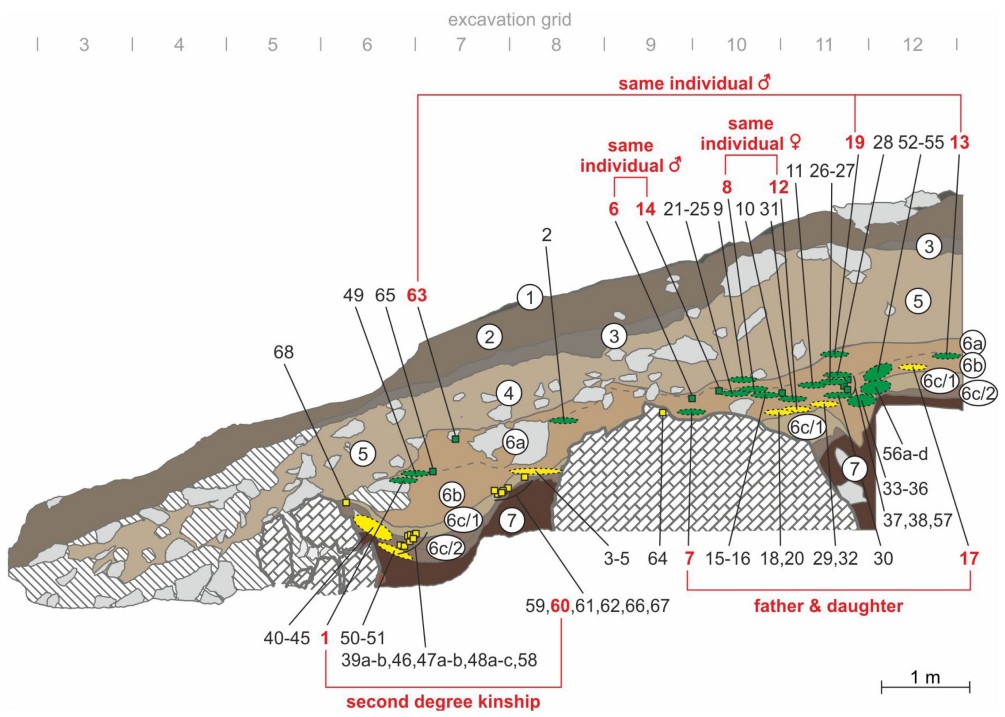

**Extended Data Fig. 2 | Plan map of Chagyrskaya Cave and locations of Neanderthal remains. A**, Spatial distribution of Neanderthal remains. The excavated area is shown in grey, and the blue line (transect A–B) marks the position of the stratigraphic profile shown in **B**. The coloured squares and ellipses denote Neanderthal remains located with exact coordinates or within the circumscribed areas, respectively, and are annotated with the corresponding fossil number(s). **B**, Stratigraphic profile along transect A–B in **A**. Locations of Neanderthal remains are projected orthogonally onto this profile, so each fossil is not necessarily shown in the stratigraphic unit from which it was recovered.

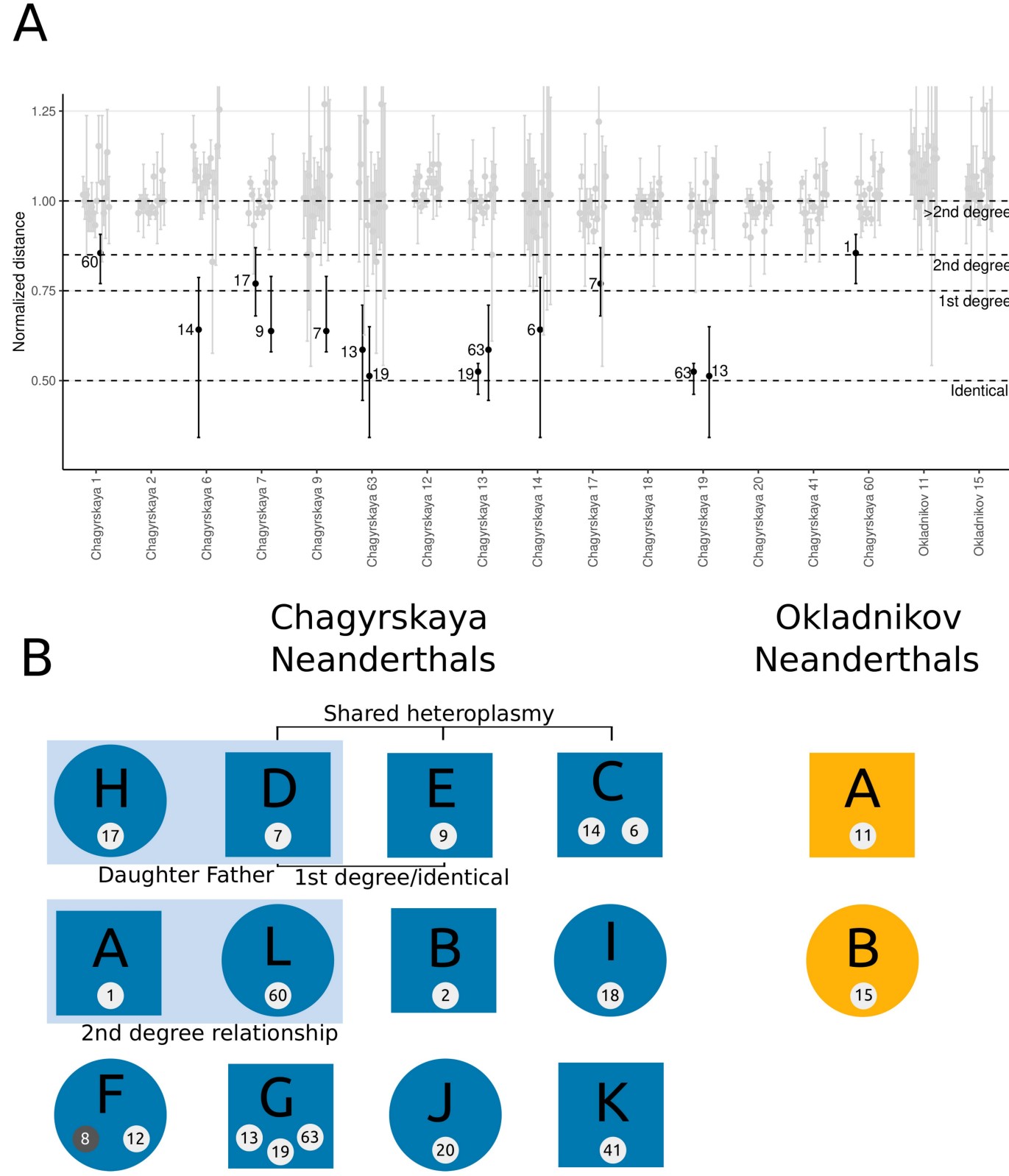

**Extended Data Fig. 3 | Normalized pairwise differences between Chagyrskaya and Okladnikov remains. A**, Points show the mean pairwise differences (y-axis) between two remains (normalized by the median difference between all pairs of remains). Remains that were identified as identical, first degree and second degree relatives are named (x-axis shows the first fossil and the number denotes the second remain). Error bars are 95% confidence intervals for 100 bootstrap estimates of the mean pairwise differences. Horizontal lines indicate the expected normalized difference for identical individuals, first degree relationships, second degree relationships and unrelated individuals[27]. **B**, Each circle/square represents an individual (blue for Chagyrskaya, orange for Okladnikov) and the small white circles indicate which remains originated from this individual. The black circle for *Chagyrskaya 8* indicates that the genomic sequence for this bone is previously published. Squares indicate that the individual is male and circles indicate that the individual is female. Individuals which are first degree relatives, second degree relatives or share heteroplasmies are marked.

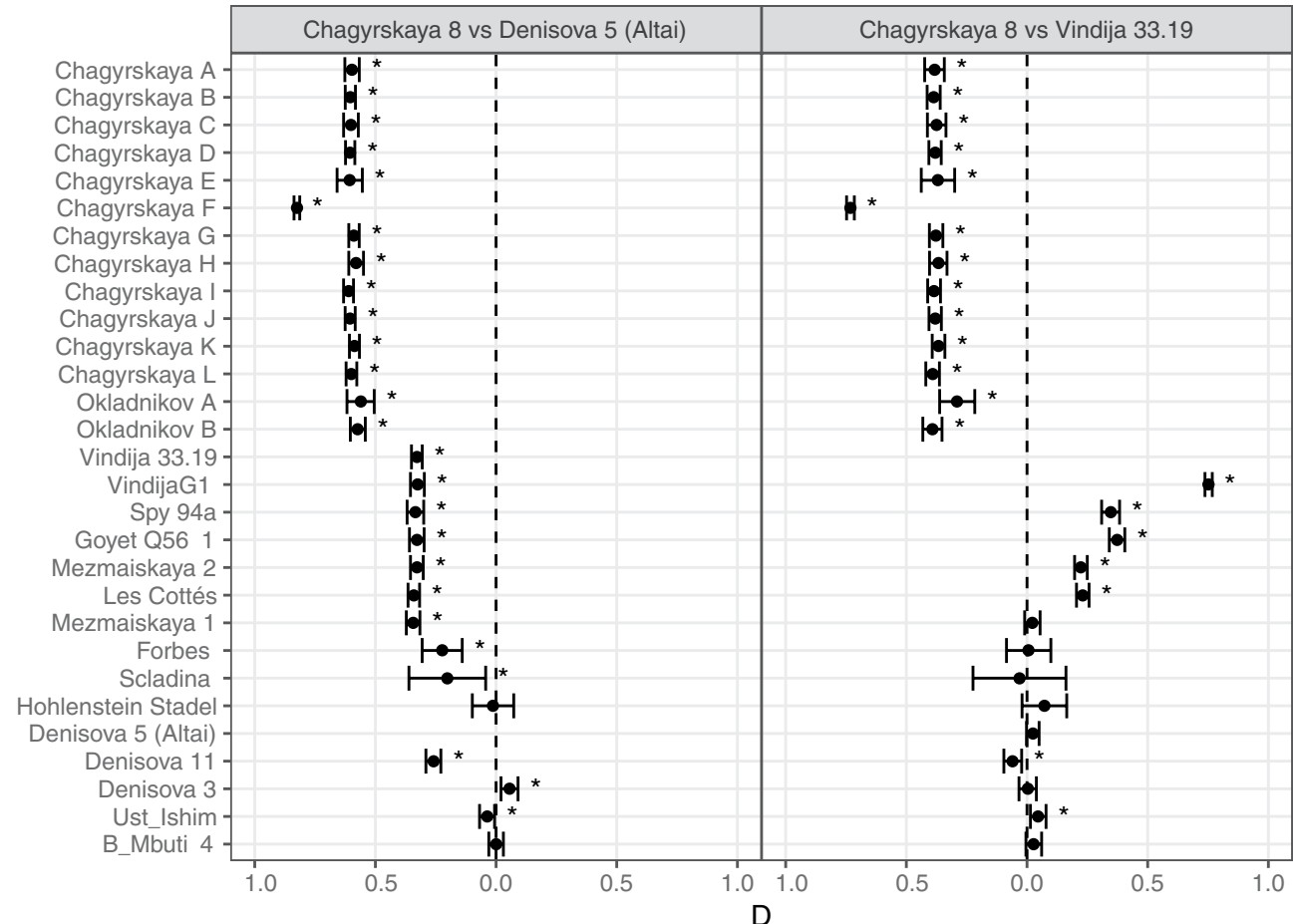

**Extended Data Fig. 4 | Sharing of variants among archaic genomes.**
The center of the errorbar show the D-statistic of the form $D((Denisova 5/Vindija33.19), Chagyrskaya 8; Test, Chimpanzee)$ and error bars are the corresponding 95% confidence intervals calculated for 643,472 SNPs using a weighted block jackknife and a block size of 5 Mb. Points with |Z-score| > 2 are annotated with an asterisk. The dashed vertical line is at D = 0. Note that *Chagyrskaya F* is the same individual as *Chagyrskaya 8* and *VindijaG1* is the same individual as *Vindija 33.19*.

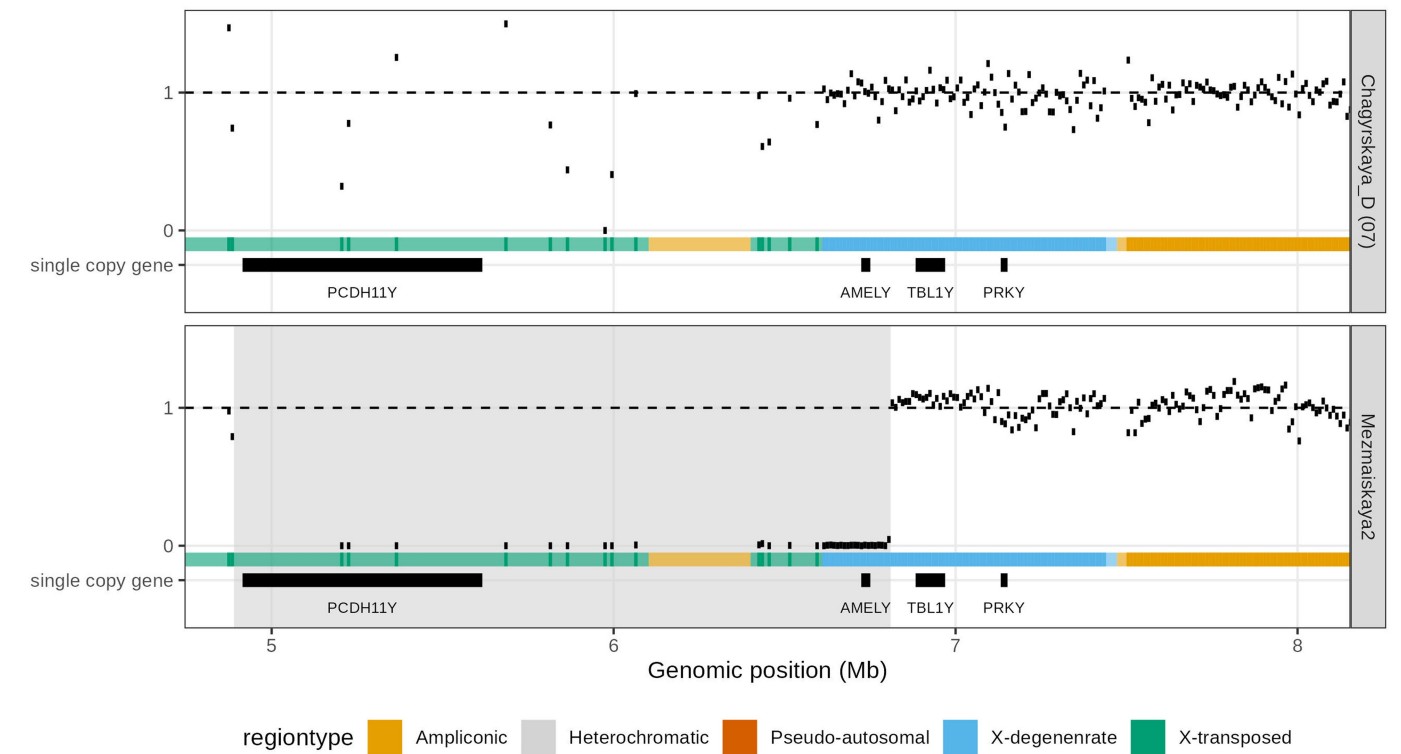

**Extended Data Fig. 5 | Deletion of the *AMELY* gene on the Y-chromosome.**
Deletion of 1.8 Mb of sequence on the Y-chromosome of *Mezmaiskaya 2* (bottom panel, light grey) compared to *Chagyrskaya D* (top panel, no deletion). The horizontal axis shows the genomic position on the Y-chromosome and the vertical axis shows the coverage in bins of 10 kb, normalized by the chromosome-wide average coverage. Bin colours indicate the region classes on the human reference Y-chromosome, with darker regions indicating coverage by the Y-chromosome capture array. Black bars denote known coding genes.

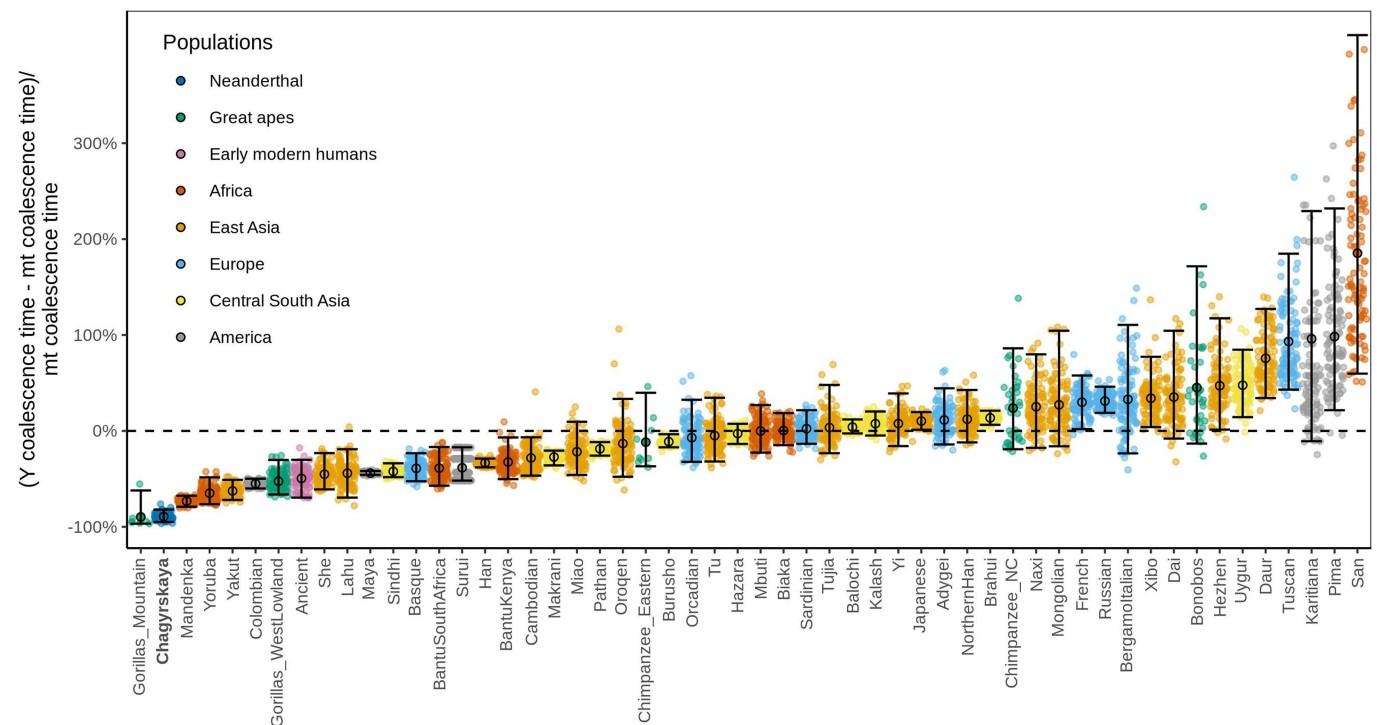

**Extended Data Fig. 6 | Ratios of mitochondrial DNA to Y-chromosome diversity.** Black circles indicate mean estimates for each population and error bars are the corresponding 95% confidence intervals using 100 bootstrap iterations. Negative values denote lower Y-chromosome diversity than mitochondrial (mt) DNA diversity.

**Extended Data Table 1 | Neanderthal remains from Chagyrskaya and Okladnikov Caves included in this study for DNA analysis or ¹⁴C dating**

| Fossil name (excavation ID) | Sample ID Leipzig | Excavation square | Strati-graphic unit / layer | Anatomical element | Ancient DNA present (a) | Radiocarbon AMS-code | Previous fossil names(s) (b) |
|---|---|---|---|---|---|---|---|
| | | | | **Chagyrskaya Cave** | | | |
| Chagyrskaya 1 | SP4815 | Л6 | 6b | lower left deciduous canine | genome capture | | |
| Chagyrskaya 2 | SP2678 | M8 | 6b | atlas fragment, child | genome capture | | |
| Chagyrskaya 6 | SP2923 | H9 | 6b | right mandible fragment with C–M2 | genome capture | | |
| Chagyrskaya 7 | SP3358 | H11 | 6c/1 | thoracal vertebral process fragment | genome capture | | |
| Chagyrskaya 8 | SP3393 | H10 | 6b | distal manual phalanx | full genome | | Chagyrskaya 8 (1) |
| Chagyrskaya 9 | SP3394 | H10 | 6a | left proximal ulna fragment | genome capture | MAMS-24965 | |
| Chagyrskaya 12 | SP4816 | M10 | 6c/1 | left P3 | genome capture | | |
| Chagyrskaya 13 | SP4817 | O12 | 6b | left I1 | genome capture | | |
| Chagyrskaya 14 | SP4818 | H10 | 6b | left I2 | genome capture | | |
| Chagyrskaya 17 | SP4819 | M12 | 6c/1 | right P4 | genome capture | | |
| Chagyrskaya 18 | SP4820 | H10, M10 | 6c/1 | left dm1 | genome capture | | |
| Chagyrskaya 19 | SP4821 | O11 | 6a | left dm2 | genome capture | | |
| Chagyrskaya 20 | SP4822 | M10 | 6c/1 | right upper? dc | genome capture | | |
| Chagyrskaya 41 | SP4823 | K6 | 6c/1 | right P3 | genome capture | | |
| Chagyrskaya 50 | SP4824 | K6 | 6c/2 | lower P3, left | no data | | |
| Chagyrskaya 56c | SP4825 | H11 | 6b | middle phalanx of the hand | no data | | |
| Chagyrskaya 60 | SP4826 | K7 | 6c/2 | manual middle phalanx (V?) | genome capture | | |
| Chagyrskaya 63 | SP4828 | И7 | 6a | left upper M2 crown | genome capture | | |
| | | | | **Okladnikov Cave** | | | |
| Okladnikov 14 (S-84/164) | SP1087 | Б1 | 3 | right humerus, distal half, child | mtDNA | OxA-X-2762-12 | Okladnikov 7 (2, 6), OK1 (3), OK2/Okladnikov 2 (4, 5) |
| | SP2984 | | | | no data | | |
| Okladnikov 1 (S-84/349) | SP2982 | Г4 | 3 | left femur, distal half, child | no data | OxA-X-2762-13 | Okladnikov 8 (2,6), OK2 (3), OK1 (4) |
| Okladnikov 11 (S-84/276) | SP2981 | Г3 | 2 | left femur, proximal half, child | genome capture | | |
| Okladnikov 15 (S-84/78) | SP2985 | В2 | 2 | adult humerus | genome capture | OxA-X-2737-20 | Okladnikov 9 (2) |
| Okladnikov 2 | SP2979 | Г4 | 3 | middle phalanx | no data | | Okladnikov 6 (2) |
| Okladnikov 5 (S-84/3-1) | SP2980 | H1 | 3 | manual middle phalanx, ray 2 | no data | | |
| Okladnikov 4 | SP2983 | Л1 | 1 | left fifth metatarsal | no data | | |
| Okladnikov 8 (S-84/3753) | SP2987 | В2 | 2 | left talus fragment | no data | | |
| Okladnikov 10 (S-84/324) | SP2986 | В4 | 2 | left talus | no data | | |
| Okladnikov 13 | SP2988 | Л1 | 1 | os parietale (Homo??) | no data | | |

All remains from Chagyrskaya Cave were recovered from lithoseries II, and all those from Okladnikov Cave were recovered from the Shelter. (**a**) This study, except for *Chagyrskaya 8*[5] and Okladnikov sample SP1087 (mtDNA only[41]); 'genome capture' indicates that mitochondrial, Y-chromosome and nuclear capture has been performed, and 'no data' indicates that no ancient DNA was detected. (**b**) Source references: 1,[5]; 2,[54]; 3,[55]; 4,[56]; 5,[41]; 6[25].

# Reporting Summary

## Statistics

For all statistical analyses, confirm that the following items are present in the figure legend, table legend, main text, or Methods section.

| n/a | Confirmed | |
|---|---|---|
| ☐ | ☒ | The exact sample size (*n*) for each experimental group/condition, given as a discrete number and unit of measurement |
| ☐ | ☒ | A statement on whether measurements were taken from distinct samples or whether the same sample was measured repeatedly |
| ☐ | ☒ | The statistical test(s) used AND whether they are one- or two-sided<br>*Only common tests should be described solely by name; describe more complex techniques in the Methods section.* |
| ☐ | ☒ | A description of all covariates tested |
| ☐ | ☒ | A description of any assumptions or corrections, such as tests of normality and adjustment for multiple comparisons |
| ☐ | ☒ | A full description of the statistical parameters including central tendency (e.g. means) or other basic estimates (e.g. regression coefficient) AND variation (e.g. standard deviation) or associated estimates of uncertainty (e.g. confidence intervals) |
| ☐ | ☒ | For null hypothesis testing, the test statistic (e.g. *F*, *t*, *r*) with confidence intervals, effect sizes, degrees of freedom and *P* value noted<br>*Give P values as exact values whenever suitable.* |
| ☐ | ☒ | For Bayesian analysis, information on the choice of priors and Markov chain Monte Carlo settings |
| ☒ | ☐ | For hierarchical and complex designs, identification of the appropriate level for tests and full reporting of outcomes |
| ☒ | ☐ | Estimates of effect sizes (e.g. Cohen's *d*, Pearson's *r*), indicating how they were calculated |

*Our web collection on statistics for biologists contains articles on many of the points above.*

## Software and code

Policy information about availability of computer code

| Data collection | No software was used for the collection of data. |
|---|---|
| Data analysis | Data was analysed using the leehom (version 1.2.16) package that is available at https://bioinf.eva.mpg.de/, bam-rmdup package that is available at https://github.com/mpieva/biohazard-tools, Python v3.7.3, BEAST v2.6.6, Tracer v1.7, Figtree v1.4.4, samtools v1.3.1-21, admixfrog v0.6.1, MAFFT version 7.453, bwa-0.7.17 version, admixtools v 7.0.2, READ (no version number available but can be download here https://bitbucket.org/tguenther/read/src/master/) and R v4.0.3 |

For manuscripts utilizing custom algorithms or software that are central to the research but not yet described in published literature, software must be made available to editors and reviewers. We strongly encourage code deposition in a community repository (e.g. GitHub). See the Nature Portfolio guidelines for submitting code & software for further information.

## Data

Policy information about availability of data

All manuscripts must include a data availability statement. This statement should provide the following information, where applicable:
- Accession codes, unique identifiers, or web links for publicly available datasets
- A description of any restrictions on data availability
- For clinical datasets or third party data, please ensure that the statement adheres to our policy

Raw data for each library are available in the European Nucleotide Archive under accession number PRJEB55327. Mapped bam files for all specimens and individuals, VCF files, consensus FASTA mtDNA sequences and a multiple alignment of all mtDNA can be downloaded from http://ftp.eva.mpg.de/neandertal/ChagyrskayaOkladnikov/.

ISOGG database (newest version is 15.73 from 11 Jul 2020) can be found at https://isogg.org/tree/index.html.
UCSC Table browser is available at http://genome.ucsc.edu.

# Field-specific reporting

Please select the one below that is the best fit for your research. If you are not sure, read the appropriate sections before making your selection.

☒ Life sciences ☐ Behavioural & social sciences ☐ Ecological, evolutionary & environmental sciences

For a reference copy of the document with all sections, see nature.com/documents/nr-reporting-summary-flat.pdf

# Life sciences study design

All studies must disclose on these points even when the disclosure is negative.

| | |
|---|---|
| Sample size | No sample size was determined in advance. We sampled all 27 available specimens from Chagyrskaya cave and Okladnikov cave - which were either tooth or bone. |
| Data exclusions | We excluded 10 specimens which had poor DNA preservation |
| Replication | For each specimen that showed evidence of DNA preservation we prepared between 2 to 14 independent DNA extracts. The preservation of ancient DNA varies between different ancient remains. In fact it is highly heterogeneous even within the same remain. Therefore not all replicates were successful. To allow the reproducibility of the analyses, all filtering steps and the comparative data used are detailed in the Methods section and the supplementary information. |
| Randomization | No randomization was performed as this was not relevant for our study. All samples were evaluated for the presence of ancient hominin DNA and analysis continued for those that contained ancient DNA. |
| Blinding | Blinding was not relevant for data collection as samples were selected from ancient human remains. |

# Behavioural & social sciences study design

All studies must disclose on these points even when the disclosure is negative.

| | |
|---|---|
| Study description | Briefly describe the study type including whether data are quantitative, qualitative, or mixed-methods (e.g. qualitative cross-sectional, quantitative experimental, mixed-methods case study). |
| Research sample | State the research sample (e.g. Harvard university undergraduates, villagers in rural India) and provide relevant demographic information (e.g. age, sex) and indicate whether the sample is representative. Provide a rationale for the study sample chosen. For studies involving existing datasets, please describe the dataset and source. |
| Sampling strategy | Describe the sampling procedure (e.g. random, snowball, stratified, convenience). Describe the statistical methods that were used to predetermine sample size OR if no sample-size calculation was performed, describe how sample sizes were chosen and provide a rationale for why these sample sizes are sufficient. For qualitative data, please indicate whether data saturation was considered, and what criteria were used to decide that no further sampling was needed. |
| Data collection | Provide details about the data collection procedure, including the instruments or devices used to record the data (e.g. pen and paper, computer, eye tracker, video or audio equipment) whether anyone was present besides the participant(s) and the researcher, and whether the researcher was blind to experimental condition and/or the study hypothesis during data collection. |
| Timing | Indicate the start and stop dates of data collection. If there is a gap between collection periods, state the dates for each sample cohort. |
| Data exclusions | If no data were excluded from the analyses, state so OR if data were excluded, provide the exact number of exclusions and the rationale behind them, indicating whether exclusion criteria were pre-established. |
| Non-participation | State how many participants dropped out/declined participation and the reason(s) given OR provide response rate OR state that no participants dropped out/declined participation. |
| Randomization | If participants were not allocated into experimental groups, state so OR describe how participants were allocated to groups, and if allocation was not random, describe how covariates were controlled. |

# Ecological, evolutionary & environmental sciences study design

All studies must disclose on these points even when the disclosure is negative.

| | |
|---|---|
| Study description | *Briefly describe the study. For quantitative data include treatment factors and interactions, design structure (e.g. factorial, nested, hierarchical), nature and number of experimental units and replicates.* |
| Research sample | *Describe the research sample (e.g. a group of tagged Passer domesticus, all Stenocereus thurberi within Organ Pipe Cactus National Monument), and provide a rationale for the sample choice. When relevant, describe the organism taxa, source, sex, age range and any manipulations. State what population the sample is meant to represent when applicable. For studies involving existing datasets, describe the data and its source.* |
| Sampling strategy | *Note the sampling procedure. Describe the statistical methods that were used to predetermine sample size OR if no sample-size calculation was performed, describe how sample sizes were chosen and provide a rationale for why these sample sizes are sufficient.* |
| Data collection | *Describe the data collection procedure, including who recorded the data and how.* |
| Timing and spatial scale | *Indicate the start and stop dates of data collection, noting the frequency and periodicity of sampling and providing a rationale for these choices. If there is a gap between collection periods, state the dates for each sample cohort. Specify the spatial scale from which the data are taken* |
| Data exclusions | *If no data were excluded from the analyses, state so OR if data were excluded, describe the exclusions and the rationale behind them, indicating whether exclusion criteria were pre-established.* |
| Reproducibility | *Describe the measures taken to verify the reproducibility of experimental findings. For each experiment, note whether any attempts to repeat the experiment failed OR state that all attempts to repeat the experiment were successful.* |
| Randomization | *Describe how samples/organisms/participants were allocated into groups. If allocation was not random, describe how covariates were controlled. If this is not relevant to your study, explain why.* |
| Blinding | *Describe the extent of blinding used during data acquisition and analysis. If blinding was not possible, describe why OR explain why blinding was not relevant to your study.* |

Did the study involve field work? ☐ Yes ☐ No

## Field work, collection and transport

| | |
|---|---|
| Field conditions | *Describe the study conditions for field work, providing relevant parameters (e.g. temperature, rainfall).* |
| Location | *State the location of the sampling or experiment, providing relevant parameters (e.g. latitude and longitude, elevation, water depth).* |
| Access & import/export | *Describe the efforts you have made to access habitats and to collect and import/export your samples in a responsible manner and in compliance with local, national and international laws, noting any permits that were obtained (give the name of the issuing authority, the date of issue, and any identifying information).* |
| Disturbance | *Describe any disturbance caused by the study and how it was minimized.* |

# Reporting for specific materials, systems and methods

We require information from authors about some types of materials, experimental systems and methods used in many studies. Here, indicate whether each material, system or method listed is relevant to your study. If you are not sure if a list item applies to your research, read the appropriate section before selecting a response.

## Materials & experimental systems

| n/a | Involved in the study |
|---|---|
| ☒ | ☐ Antibodies |
| ☒ | ☐ Eukaryotic cell lines |
| ☐ | ☒ Palaeontology and archaeology |
| ☒ | ☐ Animals and other organisms |
| ☒ | ☐ Human research participants |
| ☒ | ☐ Clinical data |
| ☒ | ☐ Dual use research of concern |

## Methods

| n/a | Involved in the study |
|---|---|
| ☒ | ☐ ChIP-seq |
| ☒ | ☐ Flow cytometry |
| ☒ | ☐ MRI-based neuroimaging |

# Antibodies

| | |
|---|---|
| Antibodies used | *Describe all antibodies used in the study; as applicable, provide supplier name, catalog number, clone name, and lot number.* |
| Validation | *Describe the validation of each primary antibody for the species and application, noting any validation statements on the manufacturer's website, relevant citations, antibody profiles in online databases, or data provided in the manuscript.* |

# Eukaryotic cell lines

Policy information about cell lines

| | |
|---|---|
| Cell line source(s) | *State the source of each cell line used.* |
| Authentication | *Describe the authentication procedures for each cell line used OR declare that none of the cell lines used were authenticated.* |
| Mycoplasma contamination | *Confirm that all cell lines tested negative for mycoplasma contamination OR describe the results of the testing for mycoplasma contamination OR declare that the cell lines were not tested for mycoplasma contamination.* |
| Commonly misidentified lines (See ICLAC register) | *Name any commonly misidentified cell lines used in the study and provide a rationale for their use.* |

# Palaeontology and Archaeology

| | |
|---|---|
| Specimen provenance | Materials were acquired as part of an agreement of scientific cooperation between the Institute of Archaeology and Ethnography, Siberian Branch of the Russian Academy of Sciences and the Max Planck Institute for Evolutionary Anthropology for projects in the field of palaeogenetics in North Asia, signed on December 25, 2018 and valid for a duration of five years. The Institute of Archaeology and Ethnography, Siberian Branch of the Russian Academy of Science oversees the excavation of Chagyrskaya Cave and Okladnikov Cave and obtained all permits necessary for conducting archaeological fieldwork and research associated with this project from the Ministry of Culture of the Russian Federation. |
| Specimen deposition | Sample material that was collected from the specimens was used up in DNA extraction and library preparation and the DNA libraries are stored at the Max Planck Institute for Evolutionary Anthropology in Leipzig, Germany. |
| Dating methods | Dating for three pieces of charcoal were performed at the Oxford Radiocarbon Accelerator Unit (ORAU), samples were prepared using an oxidation protocol (modified ABOx-SC) to or remove or reduce contamination from younger carbon, and were then measured by accelerator mass spectrometry.<br>The Neanderthal bone from Chagyrskaya cave was pretreated at the Department of Human Evolution of Max Planck Institute for Evolutionary Anthropology (MPI-EVA), Leipzig, Germany. The sample was sent to Curt Engelhorn Centre for Archaeometry (CEZA), Mannheim, Germany (Lab Code MAMS), where it was graphitized and dated.<br>For the three Neanderthal bones from Okladnikov Cave we took bone powder samples using an NSK drill kit with a tungsten carbide bit, and extracted collagen using a non-routine method consisting of decalcification using dilute HCl acid, followed by gelatinization and lyophilization. We then hydrolysed the collagen and separated the underivatized amino acids using preparative liquid chromatography (Prep-LC), employing the method described by to collect the amino acid hydroxyproline (HYP). This was then combusted using an EA-IRMS system (Carlo Erba EA1108/Europa Geo 20/20) operating in continuous-flow mode, from which we obtained the C/N atomic ratios and other analytical data. Finally, we graphitized the HYP fraction and measured it on the HVEE accelerator mass spectrometer at ORAU. |

☒ Tick this box to confirm that the raw and calibrated dates are available in the paper or in Supplementary Information.

| | |
|---|---|
| Ethics oversight | All necessary permits for excavations at Chagyrskaya Cave and Okladnikov Cave were obtained by the Institute of Archaeology and Ethnography, Siberian Branch of the Russian Academy of Science from the Ministry of Culture of the Russian Federation |

Note that full information on the approval of the study protocol must also be provided in the manuscript.

# Animals and other organisms

Policy information about studies involving animals; ARRIVE guidelines recommended for reporting animal research

| | |
|---|---|
| Laboratory animals | *For laboratory animals, report species, strain, sex and age OR state that the study did not involve laboratory animals.* |
| Wild animals | *Provide details on animals observed in or captured in the field; report species, sex and age where possible. Describe how animals were caught and transported and what happened to captive animals after the study (if killed, explain why and describe method; if released, say where and when) OR state that the study did not involve wild animals.* |
| Field-collected samples | *For laboratory work with field-collected samples, describe all relevant parameters such as housing, maintenance, temperature, photoperiod and end-of-experiment protocol OR state that the study did not involve samples collected from the field.* |
| Ethics oversight | *Identify the organization(s) that approved or provided guidance on the study protocol, OR state that no ethical approval or guidance was required and explain why not.* |

Note that full information on the approval of the study protocol must also be provided in the manuscript.

# Human research participants

Policy information about studies involving human research participants

| | |
|---|---|
| Population characteristics | *Describe the covariate-relevant population characteristics of the human research participants (e.g. age, gender, genotypic information, past and current diagnosis and treatment categories). If you filled out the behavioural & social sciences study design questions and have nothing to add here, write "See above."* |
| Recruitment | *Describe how participants were recruited. Outline any potential self-selection bias or other biases that may be present and how these are likely to impact results.* |
| Ethics oversight | *Identify the organization(s) that approved the study protocol.* |

Note that full information on the approval of the study protocol must also be provided in the manuscript.

# Clinical data

Policy information about clinical studies

All manuscripts should comply with the ICMJE guidelines for publication of clinical research and a completed CONSORT checklist must be included with all submissions.

| | |
|---|---|
| Clinical trial registration | *Provide the trial registration number from ClinicalTrials.gov or an equivalent agency.* |
| Study protocol | *Note where the full trial protocol can be accessed OR if not available, explain why.* |
| Data collection | *Describe the settings and locales of data collection, noting the time periods of recruitment and data collection.* |
| Outcomes | *Describe how you pre-defined primary and secondary outcome measures and how you assessed these measures.* |

# Dual use research of concern

Policy information about dual use research of concern

## Hazards

Could the accidental, deliberate or reckless misuse of agents or technologies generated in the work, or the application of information presented in the manuscript, pose a threat to:

| No | Yes | |
|----|-----|---|
| ☒ | ☐ | Public health |
| ☒ | ☐ | National security |
| ☒ | ☐ | Crops and/or livestock |
| ☒ | ☐ | Ecosystems |
| ☒ | ☐ | Any other significant area |

## Experiments of concern

Does the work involve any of these experiments of concern:

| No | Yes | |
|----|-----|---|
| ☒ | ☐ | Demonstrate how to render a vaccine ineffective |
| ☒ | ☐ | Confer resistance to therapeutically useful antibiotics or antiviral agents |
| ☒ | ☐ | Enhance the virulence of a pathogen or render a nonpathogen virulent |
| ☒ | ☐ | Increase transmissibility of a pathogen |
| ☒ | ☐ | Alter the host range of a pathogen |
| ☒ | ☐ | Enable evasion of diagnostic/detection modalities |
| ☒ | ☐ | Enable the weaponization of a biological agent or toxin |
| ☒ | ☐ | Any other potentially harmful combination of experiments and agents |

# ChIP-seq

## Data deposition

☐ Confirm that both raw and final processed data have been deposited in a public database such as GEO.

☐ Confirm that you have deposited or provided access to graph files (e.g. BED files) for the called peaks.

**Data access links**
*May remain private before publication.*

*For "Initial submission" or "Revised version" documents, provide reviewer access links. For your "Final submission" document, provide a link to the deposited data.*

**Files in database submission**

*Provide a list of all files available in the database submission.*

**Genome browser session**
(e.g. UCSC)

*Provide a link to an anonymized genome browser session for "Initial submission" and "Revised version" documents only, to enable peer review. Write "no longer applicable" for "Final submission" documents.*

## Methodology

**Replicates**

*Describe the experimental replicates, specifying number, type and replicate agreement.*

**Sequencing depth**

*Describe the sequencing depth for each experiment, providing the total number of reads, uniquely mapped reads, length of reads and whether they were paired- or single-end.*

**Antibodies**

*Describe the antibodies used for the ChIP-seq experiments; as applicable, provide supplier name, catalog number, clone name, and lot number.*

**Peak calling parameters**

*Specify the command line program and parameters used for read mapping and peak calling, including the ChIP, control and index files used.*

**Data quality**

*Describe the methods used to ensure data quality in full detail, including how many peaks are at FDR 5% and above 5-fold enrichment.*

**Software**

*Describe the software used to collect and analyze the ChIP-seq data. For custom code that has been deposited into a community repository, provide accession details.*

# Flow Cytometry

## Plots

Confirm that:

☐ The axis labels state the marker and fluorochrome used (e.g. CD4-FITC).

☐ The axis scales are clearly visible. Include numbers along axes only for bottom left plot of group (a 'group' is an analysis of identical markers).

☐ All plots are contour plots with outliers or pseudocolor plots.

☐ A numerical value for number of cells or percentage (with statistics) is provided.

## Methodology

**Sample preparation**

*Describe the sample preparation, detailing the biological source of the cells and any tissue processing steps used.*

**Instrument**

*Identify the instrument used for data collection, specifying make and model number.*

**Software**

*Describe the software used to collect and analyze the flow cytometry data. For custom code that has been deposited into a community repository, provide accession details.*

**Cell population abundance**

*Describe the abundance of the relevant cell populations within post-sort fractions, providing details on the purity of the samples and how it was determined.*

**Gating strategy**

*Describe the gating strategy used for all relevant experiments, specifying the preliminary FSC/SSC gates of the starting cell population, indicating where boundaries between "positive" and "negative" staining cell populations are defined.*

☐ Tick this box to confirm that a figure exemplifying the gating strategy is provided in the Supplementary Information.

# Magnetic resonance imaging

## Experimental design

**Design type**

*Indicate task or resting state; event-related or block design.*

| Design specifications | *Specify the number of blocks, trials or experimental units per session and/or subject, and specify the length of each trial or block (if trials are blocked) and interval between trials.* |
|---|---|
| Behavioral performance measures | *State number and/or type of variables recorded (e.g. correct button press, response time) and what statistics were used to establish that the subjects were performing the task as expected (e.g. mean, range, and/or standard deviation across subjects).* |

## Acquisition

| Imaging type(s) | *Specify: functional, structural, diffusion, perfusion.* |
|---|---|
| Field strength | *Specify in Tesla* |
| Sequence & imaging parameters | *Specify the pulse sequence type (gradient echo, spin echo, etc.), imaging type (EPI, spiral, etc.), field of view, matrix size, slice thickness, orientation and TE/TR/flip angle.* |
| Area of acquisition | *State whether a whole brain scan was used OR define the area of acquisition, describing how the region was determined.* |

Diffusion MRI ☐ Used ☐ Not used

## Preprocessing

| Preprocessing software | *Provide detail on software version and revision number and on specific parameters (model/functions, brain extraction, segmentation, smoothing kernel size, etc.).* |
|---|---|
| Normalization | *If data were normalized/standardized, describe the approach(es): specify linear or non-linear and define image types used for transformation OR indicate that data were not normalized and explain rationale for lack of normalization.* |
| Normalization template | *Describe the template used for normalization/transformation, specifying subject space or group standardized space (e.g. original Talairach, MNI305, ICBM152) OR indicate that the data were not normalized.* |
| Noise and artifact removal | *Describe your procedure(s) for artifact and structured noise removal, specifying motion parameters, tissue signals and physiological signals (heart rate, respiration).* |
| Volume censoring | *Define your software and/or method and criteria for volume censoring, and state the extent of such censoring.* |

## Statistical modeling & inference

| Model type and settings | *Specify type (mass univariate, multivariate, RSA, predictive, etc.) and describe essential details of the model at the first and second levels (e.g. fixed, random or mixed effects; drift or auto-correlation).* |
|---|---|
| Effect(s) tested | *Define precise effect in terms of the task or stimulus conditions instead of psychological concepts and indicate whether ANOVA or factorial designs were used.* |

Specify type of analysis: ☐ Whole brain ☐ ROI-based ☐ Both

| Statistic type for inference (See Eklund et al. 2016) | *Specify voxel-wise or cluster-wise and report all relevant parameters for cluster-wise methods.* |
|---|---|
| Correction | *Describe the type of correction and how it is obtained for multiple comparisons (e.g. FWE, FDR, permutation or Monte Carlo).* |

## Models & analysis

| n/a | Involved in the study |
|---|---|
| ☐ | ☐ Functional and/or effective connectivity |
| ☐ | ☐ Graph analysis |
| ☐ | ☐ Multivariate modeling or predictive analysis |

| Functional and/or effective connectivity | *Report the measures of dependence used and the model details (e.g. Pearson correlation, partial correlation, mutual information).* |
|---|---|
| Graph analysis | *Report the dependent variable and connectivity measure, specifying weighted graph or binarized graph, subject- or group-level, and the global and/or node summaries used (e.g. clustering coefficient, efficiency, etc.).* |
| Multivariate modeling and predictive analysis | *Specify independent variables, features extraction and dimension reduction, model, training and evaluation metrics.* |

