## [Peer Review File · Nature]

Manuscript Title: Genetic insights into the social organization of Neanderthals

Reviewer Comments & Author Rebuttals

Reviewer Reports on the Initial Version:

Referee expertise:

Referee #1: evolutionary anthropology

Referee #2: palaeogenomics

Referee #3: ancient genomics

Referee #4: Palaeolithic archaeology

Referees' comments:

Referee #1 (Remarks to the Author):

I thoroughly enjoyed reading this manuscript, which presented important new information about Neanderthal social organisation and in particular provided empirical evidence for female migration/dispersal between communities as well as relatively small community size. In a sense, the findings are not unexpected as comparison with modern humans and great apes indicates that female dispersal was likely in ancient populations. Similarly, the finding of relatively small group size was not a surprise. However, where this manuscript excels (as well as providing novel and important results) is in its empirical support for those inferences. The consideration of interaction with Denisovans, building on previous work, is also very interesting and worthy of publication to help construct a picture of more general ecology. The first degree relative pair (father-daughter) and other findings of relatedness were less important findings, as one might expect related individuals to live together at some point in their lives, and it adds less to the theoretical literature than the findings related to male philopatry. But the father-daughter pair does serve to bring these ancient humans alive, and provide us with greater detailed insight into their lives.

I am not a technical expert in the genetic and dating methods used, but the supplementary material was highly comprehensive and easy to follow. The skeletal / dental inferences were appropriate and well justified. The treatment of uncertainties and use of scenario modelling was especially strong in support of the conclusion about the consideration of reproductive skew / group size / migration pattern. The conclusions were well supported by the data, and I appreciated the level of detail in the supplementary material: the work is rigorous in its attention to detail and also in its consideration of how different bits of evidence fit together.

I have a small number of minor comments / suggestions:

Would dispersal be a more accurate term than migration?

Table 1, row related to Chagyrskaya E? - looks like a typographic error: final column should be "Possible first-degree/identical to Chagyrskaya D"?

The phrasing in the Identification of relatives section that reads "They could in principle have been a mother-son, brother-sister or father-daughter pair" seems a little bit misleading when when discovers in the next sentence or so that the male is adult and the female is adolescent. It would be better to rephrase "There are three potential male-female combinations of first degree relatives: mother-son, brother-sister or father-daughter". Also, when considering the implications of the father-daughter finding, the fact that the daughter is adolescent is interesting: was she too young to have migrated? This may be taking inference too far but fits with a general picture of dispersal of females on maturity.

In the first paragraph of SI8, the phrasing "In contrast to modern humans, chimpanzees..." is a bit misleading. Recommend deleting "In contrast to modern humans" as the sentence is clearer without it.

Also, in the supplementary, I would appreciate a fuller rationale for why particular generation times were chosen, given that there is such spatial and temporal variation in generation time.

In conclusion, the article was written clearly, rigorous in its science, well argued, and the conclusions matched the data. Appropriate credit was given to previous work, and it makes an original and significant contribution to the literature.

Referee #2 (Remarks to the Author):

In this paper Skov et al. examine the genomes of ~50kya neanderthal remains from two caves in Siberia. In particular they are able to generate genomic SNP data from 11 different individuals from the same site. This allows the authors to examine patterns of genetic relatedness within the community, identifying one first degree and one second degree pair of related individuals. They are also able to show that these were closely related communities with large runs of homozygosity that were genetically distinct from other known neanderthal populations (interestingly being closer to contemporary European ones than earlier spatially more proximate Russian ones, matching the varying material culture of the different populations). They also examine relative Y and mtDNA effective population size and show differences that are best explained by small communities with large amounts of female gene flow.

By generating sequence data for multiple remains from the same cave that appear contemporaneous, this paper is truly unique, in regards to for the first time actually being able to use genomic data to understand the genetic relationships within Neanderthal communities, and by proxy archaic hominid social systems and structures, rather than simply focusing on how different they are to, or how they have interacted with, humans like in previous studies. They also begin to

unravel population dynamics between different Neanderthal populations at different time points. This kind of insight is something that 20 years ago we might not have dreamt was possible. So I believe this is an important paper that will be of considerable interest to geneticists, anthropologist, paleontologists and the general public. This simple technical result of simply getting DNA from this many samples of this age is also a considerable achievement and should not be underestimated. This work potentially sets the stage for the examination of more Neanderthal communities in the future, which may give us unprecedented new knowledge of archaic hominid social organization and population structure.

This study clearly also involved an immense amount of lab and analytical work, all of which appears expertly done. The authors develop new custom mtDNA, Y chromosome and nuclear captures suited for their specific purpose. They also painstakingly examine all possible ways these captures could introduce bias into subsequent results (do probes overlap multiple SNPs, how do they correspond with GC content?) and use this information to inform subsequent analysis (like comparing X/A coverage to determine sex, or relatedness). They go through extensive procedures to identify and where possible control for contamination at all levels of analysis (shotgun sequencing and the different captures). The various population genetic analysis performed is applied appropriately and always tries to account of contamination and low coverage, which should ensure the results are as robust as possible. While contamination from such old remains is inevitable, the authors have done everything they can to account for this when making their conclusions.

I could quibble with the framework of how the Y/mtDNA simulations were performed and why certain parameters were chosen over others (for example why 300 generations and why the particular demic structure), but I would not ask the authors to redo any of it, as there are an infinite set of potential options one could include, and I feel the end result is largely robust. I would suggest they provide more information on what they actually did in the main text however, and in particular state what the statistical support was for their best fitting scenario, which was at best about 7% of simulations, suggesting considerable room for improvement by refinement of the model.

One other thing that might be useful is to contextualize the results of the Y/mtDNA diversity within findings from previous work in humans, in particular the body of work from the early 2010s that looked at X/A diversity to look at sex-biased demographic events (for example Keinan and Reich, MBE 2010).

Despite being positive about the paper, I do have issues with the supplementary material, which at times seemed excessively long, a little hastily put together (there are equations with symbols missing, sudden changes in typeface) and was ordered in a way that made it a slog to get through. For example, on page 77, do we really need to know the structure of the bed file? For the filtering criteria for the autosomal capture, what does “flanks (105 bases) mean? Does the First degree relationship section on page 94 really need such a pedantic explanation of how shared mtDNA might help reveal kinship relationships? This is in contrast to complex analysis like inference of admixture time that are simply described by a (incomplete) formula. What was the point of the section “TMRCA for mitochondrial sequences”? I cannot see where this was used in the actual main text?

It would help enormously if some parts of the supplementary were re-ordered, for example putting

all the contamination results in one section, mtDNA and Y chromosome phylogenetic results in one section etc, so they could be compared, rather than having to jump around. Similarly, keeping the various heteroplasmies sections together would also help. While I see some sense in the current procedural ordering (we did A, then we did B, then we did C), it is really hard to get through 100+ pages like this. There is already an issue with regard to how much such large supplementary materials are actually being read, and this kind of formatting does not help this.

Krishna Veeramah

Referee #3 (Remarks to the Author):

Skov and colleagues have produced an impactful study that provides a new window into the structure and size of Neanderthal communities. The most remarkable aspect is the density of data produced for the Chagyrskaya Cave site, a commendable feat. With these data, the authors provide estimates of relatedness, inbreeding and uniparental marker diversity for a Neanderthal population. The results serve to strengthen what has already been inferred from single genomes, mtDNA sequences and other paleoanthropological data: (1) Neanderthals lived in small communities and (2) at least some of these show female-biased dispersal. The novelty is in the level of detail the authors provide (e.g. uniparental diversity ratios, shared heteroplasmies, population-level ROH, best-fit community sizes, migration rates).

I'd be happy to recommend this for publication in Nature. I think it will attract wide-ranging academic and public interest. As a general comment I think more room needs to be given to discussion, including synthesis of previous work (genomic and otherwise) and contextual information on comparisons with the apes (see below). In parts, the main text is bogged down in methodological detail and blow-by-blow results, some of which could be relegated to supplementary. More specific comments and corrections I would like to see addressed are below -

A. Descriptions of prior work: For the uninitiated, I don't think the abstract and introduction give the reader a clear picture of the current state of knowledge. There is prior evidence of patrilocality in Neanderthals (Lalueza-Fox et al. 2011). This study is cited, but the relevant result is not.

In figure 1 map, please provide the number of sequenced individuals for each site. It is not immediately clear which other sites have yielded multi-individual samples. It would also be useful to provide summary of available mtDNA.

Is this the first discovery of 1st/2nd degree relatives in an archaic population? If so, please highlight this achievement more clearly.

The final line in the abstract seems a little misleading. This is not the earliest known example of social organization in a Neanderthal population (French footprints) nor is it the earliest from this eastern region (Inbreeding in the Altai Neanderthal genome informed on community size). Please

rephrase to make clearer where the novelty lies.

Genomic Diversity

Unclear what light and dark colours refer to in Figure 2A and 2B.

The proportion of the genome in ROH for several Chagyrskaya individuals is greater than the Altai Neanderthal who was inferred to be the offspring of a 2nd degree mating. Are the authors suggesting these are all offspring of 2nd degree relatives or are there a differences in length distributions. Please be explicit here.

Does the X chromosome show any reduction in ROH for females? Do you see any long IBD segments shared on the X for males? A comparison of autosomal versus X chromosome diversity could be useful.

The inclusion of the great apes is very useful but warrants more discussion. Firstly, why are only Gorillas included in Fig. 2? It would be useful to know how other apes compare and their community sizes. Secondly, the authors state that “the Chagyrskaya Neanderthals have the fourth-lowest ratio of Y-chromosome to mtDNA coalescence time, with only sumatra orangutans, mountain gorillas and western chimpanzees having a more extreme ratio”, starting Line 356. What is the reader meant to deduce from this? Do some apes show more female-biased dispersal than others? A quick glance at the literature suggests Bornean Orangutans show evidence of female philopatry (van Noordwijk et al. 2012) and Bonobos have some female biases, which fits with Extended Fig 6. Do Sumatran Orangutans and Mountain Gorillas show male philopatry? If these species are to be included as comparisons the authors should provide more detailed context.

Typo in the above. Sumtra ->Sumatran

Relatedness Estimates

The discovery of close relatives within the Chagyrskaya population is a novel and impactful result. However, I believe (1) the presentation of these findings could be improved and (2) more could be done to demonstrate that all avenues for resolving ambiguous relationships were exhausted.

The discussion of these findings in the main text (starting line 159) is long-winded and not always easy to follow. It could do with a concise summary of results at the start before delving into particulars. Some of this could also be moved to supplements. For example, the identification of

remains from the same individual is not of general interest and could be compressed. It also should not take four sentences (lines 202-206) to say that two individuals are father and daughter as they have a first degree relationship but do not share mtDNA.

Also, for ease of comprehension, instead of presenting normalized pairwise differences in the main text, the authors could provide the corresponding relatedness coefficients. It seems unintuitive to discuss kinship in terms of distance rather than identity and it would make the assessment of CIs easier for the reader.

The main text seems to imply that 643,472 sites were used for relatedness (line 162). However, from the supplementary it appears that only 250,785 were used (line 1998). Please be clear about this in the main text.

The large CIs around p_0 values for some Chagyrskaya pairs prevent exact relationships from being inferred (6, 14, 9 and 7). I wonder could this be improved by increasing the number of sites used? The authors are very conservative in their filtering, keeping sites only varying in the four high coverage archaics and removing sites only derived in Chagyrskaya8. Did you try additional READ runs with more relaxed site filters? If so could these be included in the supplement? For example, why not use the full dataset of archaic genomes (Figure 1B) to determine variable sites. Given that the higher coverage Chagyrskaya individuals show no close relationship with each other (B, D, F, J and K) it could be informative to include sites that vary between these individuals.

Following from this, why did the authors not attempt to detect third degree relatives? Was it simply as this is not provided in READ output? Could the authors be explicit about the reasoning in the text. They mention limitations to the “resolution of our approach” (Line 217), but it is not clear what testing was done to establish this was the best approach.

Following the READ analysis, the authors merged data from identical individuals. It would be useful to see a plot of relatedness coefficients between all pairs of Chagyrskaya and Okladnikov individuals following these merges. A heatmap would be useful to allow quick comparisons.

The relationship between Chagyrskaya C and D could do with more exploration, given their good coverages and the shared heteroplasmy (e.g. by chromosome estimates including X). The results imply they are beyond 2nd degree relatives, but if the heteroplasmy is true they cannot be much more than 4th degree. Is this what the authors mean by “close maternal relatives” (Line 215)? Please be clearer here. If true, this result also implies multiple female line transmissions within the community or the return of daughters of females who migrated. Both are interesting possibilities, but don't seem to be considered.

Table 1 typo for Chagyrskaya E? - the relationship to other individual(s) column should read Chagyrskaya D.

Change “father daughter” to “daughter father” in Extended Data Figure 3B to reflect order of individuals.

Referee #4 (Remarks to the Author):

A. Summary of the key results

The study provides ancient DNA evidence for a number of aspects of social organization of a Neanderthal population in the eastern end of their geographical distribution, in particular a small residential group size and female-biased dispersal.

B. Originality and significance: if not novel, please include reference

The results are certainly of immediate interest to researchers on Human Origins, including archaeologists, palaeoanthropologists and geneticists, and more broadly to all those interested in the origins of human behaviour. Very little indeed is known about the social organization of Neanderthals or other Middle and Late Pleistocene humans, despite its central importance. The insight into sex-biased migration and residence patterns is particularly fascinating, since this reflects people’s decisions about where and with whom to live and is influenced by kin relations, environment and men’s and women’s foraging activities. So this is an aspect of social organization which potentially gives insight into many other aspects of life. There is only one other study that I know of that addresses this (cited in the article) and it is based on much more limited DNA data and from a different region. It is also very interesting in giving insights into the size of Neanderthal social groups on different scales. This is speculated on a lot, and is important for arguments ranging from cultural learning processes to landscape impact, but such concrete evidence is rare. Ancient DNA research has focused heavily on phylogenetic implications and to a lesser extent population demography, making this a novel and valuable study.

C. Data & methodology: validity of approach, quality of data, quality of presentation

The archaeological data and methods are valid, of good quality and clearly presented. This includes the discussion of the context and dating of the Neanderthal remains.

I have only one query in this respect with regard to the description of Okladnikov Cave in the SM page 8: I miss some information which would be helpful in understanding the dating of the sampled Neandertals from this Cave. Which layer were they from? Also, what depositional processes were involved in the formation of levels 2 and 3; is admixture likely? Because all the dates for which this is specified are from Layer 3 it seems possible that some of the sampled Neanderthal remains could be younger if they derive from layer 2 (which is relevant to the argument that they could possibly have formed part of the same community as Chagyrskaya).

D. Appropriate use of statistics and treatment of uncertainties

All good.

E. Conclusions: robustness, validity, reliability

The conclusions are clear and convincing.

F. Suggested improvements: experiments, data for possible revision

None

G. References: appropriate credit to previous work?

Yes. One small comment: the study by Lalueza Fox et al. (2011, PNAS) is cited at the start of the article (in relation to community size), but I think it could be worth referring to this again in the discussion or conclusion, specifically at line 374-376 and around 387, since this study also provides evidence for female biased migration, and is relevant to understanding whether such social organization patterns were local or widespread.

<https://www.pnas.org/doi/full/10.1073/pnas.1011553108>

H. Clarity and context: lucidity of abstract/summary, appropriateness of abstract, introduction and conclusions

The article as a whole is clear. I have a few specific suggestions with regard to clarity at the sentence level, listed below.

58-60 (abstract) 'These genetic data represent the earliest known example of social organization in a Neanderthal population at the easternmost extent of their known range.' This should perhaps be rephrased because it can be read to suggest social organization was absent before, while what is meant is that this is the first information available about the way they organized themselves socially.

110-112 This sentence is unclear. It might be helpful to add 'however' to indicate that this sentence outlines a different pattern from the previous one, and also to mention which individual from Denisova Cave is being discussed (it could be read as still discussing Denisova 5).

381-383 These sentences may be confusing and might benefit from rephrasing; it reads as if the coalescence time for the y-chromosome was part of evidence for small communities, whereas from the preceding paragraphs I understood this to be primarily relevant for migration patterns.

There are also two sentences in the supplementary materials which are not clear:

Supplementary materials p. 5: 'We also identify two remains that show a second degree relationship - one in the subseries IIa and another in subseries IIa. These remains are Chagyrskaya 1 found in the colluvial sediments and Chagyrskaya 60 found in the intact sediments (SI Figure 1.2 and SI Figure 1.3).' This should read 'subseries IIa and another in subseries IIb' I assume.

SM p. 6 'Their population size was probably small, based on genetic data ((Mafessoni et al., 2020); Supplementary Sections 8 and 9) and the limited area of living space in the cave, so it may have been used mainly as a camp for the selective hunting and consumption of bison and horses, with a preference for non-adult individuals and females (Kolobova et al., 2019).' What does population size have to do with area of the living space or use as hunting camp?

Author Rebuttals to Initial Comments:

We would like to thank the reviewers for their thoughtful comments and efforts towards improving our manuscript. We address comments specific to each reviewer below and use **red** to highlight the reviewer's question, **blue** to highlight our answer and **green** to highlight changes to the manuscript/supplementary information.

In regards to the main text we have kept track changes in, so sentences which have been added/changed are highlighted in **green** and sentences which have been moved to the supplement/deleted are highlighted in **red**.

Referee #1

I thoroughly enjoyed reading this manuscript, which presented important new information about Neanderthal social organisation and in particular provided empirical evidence for female migration/dispersal between communities as well as relatively small community size. In a sense, the findings are not unexpected as comparison with modern humans and great apes indicates that female dispersal was likely in ancient populations. Similarly, the finding of relatively small group size was not a surprise. However, where this manuscript excels (as well as providing novel and important results) is in its empirical support for those inferences. The consideration of interaction with Denisovans, building on previous work, is also very interesting and worthy of publication to help construct a picture of more general ecology. The first degree relative pair (father-daughter) and other findings of relatedness were less important findings, as one might expect related individuals to live together at some point in their lives, and it adds less to the theoretical literature than the findings related to male philopatry. But the father-daughter pair does serve to bring these ancient humans alive, and provide us with greater detailed insight into their lives.

I am not a technical expert in the genetic and dating methods used, but the supplementary material was highly comprehensive and easy to follow. The skeletal / dental inferences were appropriate and well justified. The treatment of uncertainties and use of scenario modelling was especially strong in support of the conclusion about the consideration of reproductive skew / group size / migration pattern. The conclusions were well supported by the data, and I appreciated the level of detail in the supplementary material: the work is rigorous in its attention to detail and also in its consideration of how different bits of evidence fit together.

We thank the reviewer for their insightful comments and attention to detail. We are happy that the reviewer enjoyed reading the manuscript and hope they will find the new additions helpful.

I have a small number of minor comments / suggestions:

- Would dispersal be a more accurate term than migration?

Although both 'migration' and 'dispersal' are commonly used in the literature, the definitions of these terms vary between population genetics, archaeology and behavioral ecology. We have chosen here to refer to any type of movement between populations and communities as 'migration' for consistency with a previous study of a Chagyrskaya Neanderthal (Mafessoni et al 2020).

- Table 1, row related to Chagyrskaya E? - looks like a typographic error: final column should be "Possible first-degree/identical to Chagyrskaya D"?

Thank you for spotting this. We have corrected this typo.

- The phrasing in the Identification of relatives section that reads "They could in principle

have been a mother-son, brother-sister or father-daughter pair" seems a little bit misleading when when discovers in the next sentence or so that the male is adult and the female is adolescent. It would be better to rephrase "There are three potential male-female combinations of first degree relatives: mother-son, brother-sister or father-daughter".

To address this comment, and similar comments from other reviewers, we have revised this section extensively and have modified the sentence as suggested by the reviewer.

Also, when considering the implications of the father -daughter finding, the fact that the daughter is adolescent is interesting: was she too young to have migrated? This may be taking inference too far but fits with a general picture of dispersal of females on maturity.

Though this is a really interesting possibility, the best fitting scenarios in our simulations have female migration rates between 60%-100%. We therefore cannot be certain that all females migrated, and are hesitant to speculate based on a single observation.

- In the first paragraph of SI8, the phrasing "In contrast to modern humans, chimpanzees..." is a bit misleading. Recommend deleting "In contrast to modern humans" as the sentence is clearer without it.

We agree and have removed this.

- Also, in the supplementary, I would appreciate a fuller rationale for why particular generation times were chosen, given that there is such spatial and temporal variation in generation time.

We agree that there is indeed variation in generation time in human populations with estimates of 26–30 years the past 50,000 years (Moorjani et al 2016) and 26.9 years across the past 250,000 years (Wang et al., 2021)(<https://www.biorxiv.org/content/10.1101/2021.09.07.459333v1.full>). Studies on chimpanzee has shown that the generation time varies between 23 - 31 years with a mean of 25 years (Langergraber et al. 2012).

Since we do not know the precise generation time for Neandertals, we use 29 years (Fenner 2005) throughout to be consistent with previous literature on Neandertals (eg: Prufer 2017).

We have now added a citation to (Fenner 2005) whenever we state the generation time used for analysis.

In conclusion, the article was written clearly, rigorous in its science, well argued, and the conclusions matched the data. Appropriate credit was given to previous work, and it makes an original and significant contribution to the literature.

Referee #2

In this paper Skov et al. examine the genomes of ~50kya neanderthal remains from two caves in Siberia. In particular they are able to generate genomic SNP data from 11 different individuals from the same site. This allows the authors to examine patterns of genetic relatedness within the community, identifying one first degree and one second degree pair of related individuals. They are also able to show that these were closely related communities with large runs of homozygosity that were genetically distinct from other known neanderthal populations (interestingly being closer to contemporary European ones than earlier spatially more proximate Russian ones, matching the varying material culture of the different populations). They also examine relative Y and mtDNA effective population size and show differences that are best explained by small communities with large amounts of female gene flow.

By generating sequence data for multiple remains from the same cave that appear contemporaneous, this paper is truly unique, in regards to for the first time actually being able to use genomic data to understand the genetic relationships within Neanderthal communities, and by proxy archaic hominid social systems and structures, rather than simply focusing on how different they are to, or how they have interacted with, humans like in previous studies. They also begin to unravel population dynamics between different Neanderthal populations at different time points. This kind of insight is something that 20 years ago we might not have dreamt was possible. So I believe this is an important paper that will be of considerable interest to geneticists, anthropologist, paleontologists and the general public. This simple technical result of simply getting DNA from this many samples of this age is also a considerable achievement and should not be underestimated. This work potentially sets the stage for the examination of more Neanderthal communities in the future, which may give us unprecedented new knowledge of archaic hominid social organization and population structure.

This study clearly also involved an immense amount of lab and analytical work, all of which appears expertly done. The authors develop new custom mtDNA, Y chromosome and nuclear captures suited for their specific purpose. They also painstakingly examine all possible ways these captures could introduce bias into subsequent results (do probes overlap multiple SNPs, how do they correspond with GC content?) and use this information to inform subsequent analysis (like comparing X/A coverage to determine sex, or relatedness). They go through extensive procedures to identify and where possible control for contamination at all levels of analysis (shotgun sequencing and the different captures). The various population genetic analysis performed is applied appropriately and always tries to account of contamination and low coverage, which should ensure the results are as robust as possible. While contamination from such old remains is inevitable, the authors have done everything they can to account for this when making their conclusions.

We thank the reviewer for their insightful comments and reading the supplementary material in such detail. We address the comments below;

I could quibble with the framework of how the Y/mtDNA simulations were performed and why

certain parameters were chosen over others (for example why 300 generations and why the particular demic structure), but I would not ask the authors to redo any of it, as there are an infinite set of potential options one could include, and I feel the end result is largely robust.

I would suggest they provide more information on what they actually did in the main text

however, and in particular state what the statistical support was for their best fitting scenario, which was at best about 7% of simulations, suggesting considerable room for improvement by refinement of the model.

Since the fit is an approximation of a likelihood, we would not expect 100% of simulations to fit. In a sense, likelihoods are not particularly useful on their own but only when compared to other likelihoods. We now compare the models in a more conventional way using the Akaike information criterion (AIC) - and we show the results in **SI Table 8.4**.

We further show in **SI Figure 8.3** how the relative likelihood changes with population size.

We added a section providing a brief summary of the methodology to the main text, and in the interests of space included additional detail in the Supplementary material

We approximated the likelihood of each scenario using simulations, as the proportion of simulated data sets that are within the 95%-confidence intervals of the observed data. We then used Akaike information criterion (AIC) to rank different scenarios (**SI Table 8.4**).

The best-fitting scenarios (AIC=6.2) assumed a community size of 20 individuals, with 60–100% of the females being migrants from another community (**SI Table 8.3**). Scenarios that include only skewed offspring distributions explain the data less well (AIC=7.4), and require large community sizes of 300 individuals. Scenarios with both skewed offspring distributions and female migrations does not improve the fit (AIC=8.5) obtained by assuming migration-bias alone. Scenarios that include only differences in generation time fit the data poorly (AIC=8.5) and require parameter settings that seem unrealistic (e.g. females would need to be on average twice as old as males, **SI Table 8.3**). Previous estimates of Neanderthal community sizes range from 3 to 60 individuals^{5,16,17,19} and in this range the best fitting scenarios include female migration (**SI Figure 8.3**). This result suggests that female-biased migration was a major factor in the social organization of the Chagyrskaya Neanderthal community.

One other thing that might be useful is to contextualize the results of the Y/mtDNA diversity within findings from previous work in humans, in particular the body of work from the early 2010s that looked at X/A diversity to look at sex-biased demographic events (for example Keinan and Reich, MBE 2010).

We completely agree that this is an interesting avenue to explore and we initially did investigate the X/A-diversity. However due to scarcity of data and ascertainment bias due to our capture array the results are inconclusive (**SI section 10**).

In order to keep the manuscript as concise as possible, we chose not to present these analyses in the main text. We did however attach the 4 pages to the end of the supplementary information (**SI section 10**) since two reviewers asked about the analysis and we believe it might be of interest to some readers.

Despite being positive about the paper, I do have issues with the supplementary material, which at times seemed excessively long, a little hastily put together (there are equations with symbols missing, sudden changes in typeface) and was ordered in a way that made it a slog to get through.

We had initially converted from google doc to word which unfortunately seems to have messed up some equations. We now convert to pdf which keeps equations intact.

For example, on page 77, do we really need to know the structure of the bed file?

For the filtering criteria for the autosomal capture, what does “flanks (105 bases) mean?”

Does the First degree relationship section on page 94 really need such a pedantic explanation of how shared mtDNA might help reveal kinship relationships? This is in contrast to complex analysis like inference of admixture time that are simply described by a (incomplete) formula. What was the point of the section “TMRCA for mitochondrial sequences”? I cannot see where this was used in the actual main text?

It would help enormously if some parts of the supplementary were re-ordered, for example putting all the contamination results in one section, mtDNA and Y chromosome phylogenetic results in one section etc, so they could be compared, rather than having to jump around. Similarly, keeping the various heteroplasmies sections together would also help. While I see some sense in the current procedural ordering (we did A, then we did B, then we did C), it is really hard to get through 100+ pages like this. There is already an issue with regard to how much such large supplementary materials are actually being read, and this kind of formatting does not help this.

We acknowledge that the supplementary material is quite extensive in order to document all the analyses fully. In our view, this causes some tensions between reviewers (since reviewing a 100+ page supplement is a very big ask), and other readers, who appreciate having the details of the work attached to it (see also comments to that note by reviewer 1). We have therefore provided a detailed table of contents as well as references to relevant supplementary sections in the main text, and assume that interested readers will be willing to search for analyses of interest.

Krishna Veeramah

Referee #3

Skov and colleagues have produced an impactful study that provides a new window into the structure and size of Neanderthal communities. The most remarkable aspect is the density of data produced for the Chagyrskaya Cave site, a commendable feat. With these data, the authors provide estimates of relatedness, inbreeding and uniparental marker diversity for a Neanderthal population. The results serve to strengthen what has already been inferred from single genomes, mtDNA sequences and other paleoanthropological data: (1) Neanderthals lived in small communities and (2) at least some of these show female-biased dispersal. The novelty is in the level of detail the authors provide (e.g. uniparental diversity ratios, shared heteroplasmies, population-level ROH, best-fit community sizes, migration rates).

I'd be happy to recommend this for publication in Nature. I think it will attract wide-ranging academic and public interest. As a general comment I think more room needs to be given to discussion, including synthesis of previous work (genomic and otherwise) and contextual information on comparisons with the apes (see below). In parts, the main text is bogged down in methodological detail and blow-by-blow results, some of which could be relegated to supplementary. More specific comments and corrections I would like to see addressed are below -

We thank the reviewer for their helpful comments and many good suggestions. We have attempted to shorten down the manuscript and move much of the methodological detail to the supplementary section.

- A. Descriptions of prior work: For the uninitiated, I don't think the abstract and introduction give the reader a clear picture of the current state of knowledge. There is prior evidence of patrilocality in Neanderthals (Lalueza-Fox et al. 2011). This study is cited, but the relevant result is not.

It is true that patrilocality has previously been suggested (Lalueza-Fox et al. 2011). In this study the authors have partial mitochondrial sequences (with 3 positions that differ between mt haplotypes) from 6 adult individuals (3 male and 3 female). Because all the females carry different mitochondrial haplotypes while the male carry identical haplotypes they speculate that their data could be explained by a social organization such as patrilocality. However, as others (Vigilant and Langergraber 2011) have pointed out there are issues with this paper.

1) Even if these individuals are contemporaneous we cannot know if the females are from outside the community and the males are from within the community. Even if female migration did happen, how can we know that we have sampled migrants and not the descendents of migrants?

2) The authors never test the null hypothesis of no movement between groups. How many males and females would share mitochondrial haplotypes if there was no migration?

3) The authors are not able to contrast the mitochondria diversity with Y chromosome diversity - due to a lack of data. When patrilocality is discussed what is meant is a type of social organization where women move between communities but men stay. Since there is no Y chromosome diversity the authors can only claim (at most) that women are moving but we do not know anything about the men. Again some test of the null model (which is a scenario without movement) is needed to even claim that women are moving.

4) The authors assume that because the hypervariable regions are identical that means that the rest of the mitochondria is also identical. This is not a good assumption as we show in SI Table 3.9. For instance if only the hypervariable regions were used one would think there were 2 haplotypes for individuals Chagyrskaya A,B,F,G,H and L. However using the entire mitochondria there are 4.

We have modified the introduction to more fully describe the previous results and the debate it generated. The paragraph now reads

Based on fossilized footprints^{17,18}, and spatial patterns of site use¹⁹, previous studies on the social organization of Neanderthal communities have suggested that Neanderthals likely lived in small communities. In addition, partial mitochondrial DNA sequences from six adult Neandertals have been used to suggest that Neandertals may have been patrilocal¹⁶, though this has been debated²⁰.

- In figure 1 map, please provide the number of sequenced individuals for each site. It is not immediately clear which other sites have yielded multi-individual samples. It would also be useful to provide summary of available mtDNA.

We have now added the number of sequenced individuals for each site in Figure 1a and include a table with all available Neandertal mtDNAs in SI Table 3.6.

- Is this the first discovery of 1st/2nd degree relatives in an archaic population? If so, please highlight this achievement more clearly.

This is indeed the first discovery of 1st/2nd degree relationships and we have now added this to the conclusion.

For the first time, we document familial relationships between Neandertals, including a father-and-daughter pair.

- The final line in the abstract seems a little misleading. This is not the earliest known example of social organization in a Neanderthal population (French footprints) nor is it the earliest from this eastern region (Inbreeding in the Altai Neanderthal genome informed on community size). Please rephrase to make clearer where the novelty lies.

We agree and have modified the final line of the abstract to make this point more clearly:

Thus, the genetic data presented here provide a detailed documentation of the social organization of an isolated Neanderthal community at the easternmost extent of their known range.

Genomic Diversity

- Unclear what light and dark colours refer to in Figure 2A and 2B.

The dark colors are shared between figure 2a and 2b and between 2c and 2d. We have now made separate legends for each subfigure to make this more clear.

- The proportion of the genome in ROH for several Chagyrskaya individuals is greater than the Altai Neanderthal who was inferred to be the offspring of a 2nd degree mating. Are the authors suggesting these are all offspring of 2nd degree relatives or are there a differences in length distributions. Please be explicit here.

We changed this sentence to offer a more concrete interpretation of this data:

“Extensive homozygosity in the genome of Denisova 5² was used to infer that she was from a small community and that her parents were closely related. Given we find similar amounts of homozygosity in all Chagyrskaya individuals, this may have been a common feature of these Eastern Neandertals.”

- Does the X chromosome show any reduction in ROH for females? Do you see any long IBD segments shared on the X for males? A comparison of autosomal versus X chromosome diversity could be useful.

We did try to investigate the X versus autosomal diversity in the early stages of our analysis (see the newly added **SI section 10**). However, we found that technical difficulties due to

the ascertainment bias of sites and low-coverage data made these analyses inconclusive - and we chose not to include it in the main text.

With regards to the reduction in ROH for females on the X chromosome the data is also not very conclusive (See ResponseTable1 below). The amount of the genome in ROH varies between 15-30% on the autosomes and between 5-35% on the X chromosome.

Individual	Genome	cM	Percent	Segments
Chagyrskaya F	Autso	565	15.69%	81
Chagyrskaya F	Xchrom	71.8	36.99%	3
Chagyrskaya I	Autso	559	15.52%	108
Chagyrskaya I	Xchrom	10.2	5.26%	2
Chagyrskaya J	Autso	1063	29.52%	153
Chagyrskaya J	Xchrom	69.7	35.91%	7
Chagyrskaya L	Autso	711	19.74%	121
Chagyrskaya L	Xchrom	25	12.88%	5

ResponseTable1. The total length of ROH which are longer than 2.5cM for the autosomes and X chromosome is shown for 4 females with coverage>1X.

We have also added a plot with the position of ROH > 10cM in the genome for the Chagyrskaya individuals for visual inspection (**SI Figure 9.1**)

- The inclusion of the great apes is very useful but warrants more discussion. It would be useful to know how other apes compare and their community sizes.

In the main text figures we focus on comparison to Gorilla for both visualization and data availability reasons:

- Gorillas are a suitable comparison because they show similar diversity patterns to the Chagyrskaya Neandertals and the Gorillas from (Xue et al 2015) are not born in

captivity as is the case for other great apes from which we have genetic data (Prado-Martinez 2013).

- For the homozygosity (Fig 2B), the gorilla values are taken from the literature (ref (Xue et al 2015)). We make this now more explicit by adding a citation to ref (Xue et al 2015) to the legend of Figure 2.
- For the mean coalescence times, we provide data including all great ape populations in SI Figure 8.2. Figure 2d is restricted to a comparison with gorilla in order to see more clearly the Chagyrskaya individuals and refer readers to Figure S8.2.

Secondly, the authors state that “the Chagyrskaya Neanderthals have the fourth-lowest ratio of Y-chromosome to mtDNA coalescence time, with only Sumatra orangutans, mountain gorillas and western chimpanzees having a more extreme ratio”, starting Line 356. What is the reader meant to deduce from this? Do some apes show more female-biased dispersal than others? A quick glance at the literature suggests Bornean Orangutans show evidence of female philopatry (van Noordwijk et al. 2012) and Bonobos have some female biases, which fits with Extended Fig 6. Do Sumatran Orangutans and Mountain Gorillas show male philopatry? If these species are to be included as comparisons the authors should provide more detailed context.

The point we make here is that the Chagyrskaya Neandertals have among the most skewed Y-to-mt diversity ratios, which may suggest that their social organization is similar to the primates with similar ratios. We've modified the main text to make this clearer.

“In a comparison to 47 modern human populations and 10 great ape subspecies, the Chagyrskaya Neanderthals have among the lowest ratios of Y-chromosome to mtDNA coalescence time, with only Sumatra orangutans, mountain gorillas and western chimpanzees having a more extreme ratio (Extended Data Figure 6).”

- Typo in the above. Sumtra ->Sumatran

Has been changed

Relatedness Estimates

The discovery of close relatives within the Chagyrskaya population is a novel and impactful result. However, I believe (1) the presentation of these findings could be improved and (2) more could be done to demonstrate that all avenues for resolving ambiguous relationships were exhausted.

- The discussion of these findings in the main text (starting line 159) is long-winded and not always easy to follow. It could do with a concise summary of results at the start before delving into particulars. Some of this could also be moved to supplements. For example, the identification of remains from the same individual is not of general interest and could be compressed. It also should not take four sentences (lines 202-206) to say that two

individuals are father and daughter as they have a first degree relationship but do not share mtDNA.

We have now made substantial changes to the identification of remains from the same individuals and relatedness analysis.

However we haven't moved the entire section to the supplement because we believe there are some useful lessons to be learned:

- 1) Finding a deciduous and a permanent tooth would typically be regarded as being from two different individuals - but here we show that they are in fact from the same individual.
- 2) The use of mitochondrial heteroplasmies is not a common practice in the field of ancient DNA - but here we show that it can provide an extra line of evidence in determining close relationships.

- Also, for ease of comprehension, instead of presenting normalized pairwise differences in the main text, the authors could provide the corresponding relatedness coefficients. It seems unintuitive to discuss kinship in terms of distance rather than identity and it would make the assessment of CIs easier for the reader.

The presentation of our kinship in terms of p_0 is the direct output of the software used (READ)(Kuhn et al 2018)). READ is widely used for the analysis of kinship in low-coverage ancient DNA for which no reference allele frequencies are available. This makes CIs easier to evaluate for the reader, and it also allows us seamless comparisons between the two caves.

- The main text seems to imply that 643,472 sites were used for relatedness (line 162). However, from the supplementary it appears that only 250,785 were used (line 1998). Please be clear about this in the main text.

The reviewer is correct. We have modified the text to explain that we use the subset of sites that are variable among the high-coverage archaic genomes, and excluding sites private to Chagyrskaya 8. The text now reads:

"To determine if any of the remains originated from related individuals we computed the nuclear DNA divergence between the 17 remains by randomly sampling one allele from the 250,785 sites in the capture array that were variable in the high-coverage archaics (excluding private variants in *Chagyrskaya 8*, SI 5)"

- The large CIs around p_0 values for some Chagyrskaya pairs prevent exact relationships

from being inferred (6, 14, 9 and 7). I wonder could this be improved by increasing the number of sites used?

The authors are very conservative in their filtering, keeping sites only varying in the four high coverage archaics and removing sites only derived in Chagyrskaya8. Did you try additional READ runs with more relaxed site filters? If so could these be included in the supplement? For example, why not use the full dataset of archaic genomes (Figure 1B) to determine variable sites. Given that the higher coverage Chagyrskaya individuals show no close relationship with each other (B, D, F, J and K) it could be informative to include sites that vary between these individuals.

This is a great suggestion and we did start this analysis with all sites, but unfortunately found that this resulted in a coverage-specific ascertainment - see figure below in ReponseFig1. This is due to the fact that not all sites are captured equally well.

ReponseFig1. The x-axis show the number of overlapping sites for a pair of individuals and the y axis shows the normalized pairwise distance (p_0). We show the first and second degree relationships we find.

- Following from this, why did the authors not attempt to detect third degree relatives? Was it simply as this is not provided in READ output? Could the authors be explicit about the reasoning in the text. They mention limitations to the “resolution of our approach” (Line 217), but it is not clear what testing was done to establish this was the best approach.

We have only attempted to classify up to second degree relatives as this is what is supported in READ (Kuhn et al 2018). In their paper the authors write:

“We do not aim to classify higher degrees than second degree and, therefore, consider all relationships of third degree or higher as ‘unrelated’. This is a decision to keep the approach conservative and to allow for some variation within the group of unrelated individuals”

- Following the READ analysis, the authors merged data from identical individuals. It would be useful to see a plot of relatedness coefficients between all pairs of Chagyrskaya and Okladnikov individuals following these merges. A heatmap would be useful to allow quick comparisons.

We prefer the presentation with error bars because the confidence intervals are extremely important. However we think a heatmap is a nice additional visual aid and we have added the confidence intervals as text. We have created the heatmap and added it to the supplementary material as **SI Figure 7.2**.

- The relationship between Chagyrskaya C and D could do with more exploration, given their good coverages and the shared heteroplasmy (e.g. by chromosome estimates including X). The results imply they are beyond 2nd degree relatives, but if the heteroplasmy is true they cannot be much more than 4th degree.

Is this what the authors mean by “close maternal relatives” (Line 215)? Please be clearer here.

Their coverages for Chagyrskaya D is 5X and for Chagyrskaya C it is 0.17X. While it is difficult to determine their relationship exactly, we agree that a 4th degree relationship is likely. The main text now reads:

In addition, his mtDNA was identical to that of two other males, *Chagyrskaya C* and *Chagyrskaya E* (**SI Table 3.2**), including a shared mtDNA heteroplasmy at position 545 (G>A) with a frequency of A of 42-54% for *Chagyrskaya D*, 20-41% for *Chagyrskaya E* and 23-30% for *Chagyrskaya C*. Thus, they were likely close maternal relatives (for example they could have shared a grandmother and thus might have been fourth degree relatives). However, the extent of the relationship between *Chagyrskaya C* and *Chagyrskaya D* is beyond the resolution of our approach ($p_0=1.05$, 95% CI 0.94-1.16)

If true, this result also implies multiple female line transmissions within the community or the return of daughters of females who migrated. Both are interesting possibilities, but don't seem to be considered.

We agree that these are interesting considerations, but do not think our data can support such strong conclusions, given that

- we do not know at which age individuals migrate
- our simulations between 60% and 100% female migration rate fit very similarly to the data, which is consistent (but not definitive support) with multiple female lines of transmission, or return of daughters

- Table 1 typo for Chagyrskaya E? - the relationship to other individual(s) column should read Chagyrskaya D.

Has been changed.

- Change "father daughter" to "daughter father" in Extended Data Figure 3B to reflect order of individuals.

Has been updated

Referee #4

A. Summary of the key results

The study provides ancient DNA evidence for a number of aspects of social organization of a Neanderthal population in the eastern end of their geographical distribution, in particular a small residential group size and female-biased dispersal.

B. Originality and significance: if not novel, please include reference

The results are certainly of immediate interest to researchers on Human Origins, including archaeologists, palaeoanthropologists and geneticists, and more broadly to all those interested in the origins of human behaviour. Very little indeed is known about the social organization of Neanderthals or other Middle and Late Pleistocene humans, despite its central importance. The insight into sex-biased migration and residence patterns is particularly fascinating, since this reflects people's decisions about where and with whom to live and is influenced by kin relations, environment and men's and women's foraging activities. So this is an aspect of social organization which potentially gives insight into many other aspects of life. There is only one other study that I know of that addresses this (cited in the article) and it is based on much more limited DNA data and from a different region. It is also very interesting in giving insights into the size of Neanderthal social groups on different scales. This is speculated on a lot, and is important for arguments ranging from cultural learning processes to landscape impact, but such concrete evidence is rare. Ancient DNA research has focused heavily on phylogenetic implications and to a lesser extent population demography, making this is a novel and valuable study.

We thank the reviewer for the helpful comments and we respond to the comments below.

C. Data & methodology: validity of approach, quality of data, quality of presentation

The archaeological data and methods are valid, of good quality and clearly presented. This includes the discussion of the context and dating of the Neanderthal remains.

I have only one query in this respect with regard to the description of Okladnikov Cave in the SM page 8: I miss some information which would be helpful in understanding the dating of the sampled Neandertals from this Cave. Which layer were they from? Also, what depositional processes were involved in the formation of levels 2 and 3; is admixture likely? Because all the dates for which this is specified are from Layer 3 it seems possible that some of the sampled Neanderthal remains could be younger if they derive from layer 2 (which is relevant to the argument that they could possibly have formed part of the same community as Chagyrskaya).

Location details of the sequenced and/or directly dated Neanderthal remains are summarized in Extended Data Table 1. Okladnikov 11 and 15 were recovered from layer 2, and Okladnikov 1 and 14 (the same specimen as Okladnikov 2 in Skoglund et al., 2014 and Vernot et al., 2021) were retrieved from layer 3 (all from inside the Shelter). Given the uncertainties about the reliability of ¹⁴C ages published previously for Neanderthal remains from this site, we re-dated three of these specimens (Okladnikov 1, 11 and 15) using a hydroxyproline-based single amino acid approach; infinite ages (>40 ka) were obtained for all three specimens.

As regards the Shelter deposits, layers 2 and 3 were deposited by a combination of wind, water, gravity, and biotic processes. They are intercalated (see SI Figure 1.4 B) and were likely disturbed by the post-depositional activities of cave fauna, such as hyenas and rodents. We cannot, therefore, be specific about the exact ages of the Neanderthal remains at Okladnikov Cave based on the ^{14}C chronology, but the close genetic similarities of Okladnikov 11 (layer 2), 14 (layer 3) and 15 (layer 2) to the Chagyrskaya Neanderthals (who occupied that site sometime between 59 and 51 ka) implies that the two groups lived at around the same time. We have now added the following details (shown below in blue) to SI section 1, where we describe the stratigraphic sequence in Okladnikov Cave:

Both layers consist of redeposited colluvial sediments washed through the Grotto, aeolian materials deposited directly in the Shelter, limestone debris from the cave roof and walls, and zoogenic and anthropogenic components. The two layers are intercalated in the Shelter (SI Figure 1.4 B), due in part to post-depositional disturbance by cave fauna (e.g., hyena and rodent activities). The sequenced and/or directly dated Neanderthal remains from layer 2 (Okladnikov 11 and 15) and layer 3 (Okladnikov 1 and 14) (Extended Data Table 1) might therefore have similar ages or, if layer 3 was deposited much earlier than layer 2, represent a mixed-age assemblage. We consider it more likely that these individuals are broadly contemporaneous, because (a) the hydroxyproline-based ^{14}C ages reported here for Okladnikov 1, 11 and 15 are all >40 ka (see below), and (b) the Neanderthals who occupied Chagyrskaya Cave (for a short period sometime between 59 and 51 ka) and are closely related genetically to Okladnikov 11, 14 and 15 (Figure 1 C, where they are labelled Okladnikov A, 2 and B, respectively), which implies that the two sites were inhabited at around the same time (i.e., within a few millennia of each other).

D. Appropriate use of statistics and treatment of uncertainties

All good.

E. Conclusions: robustness, validity, reliability

The conclusions are clear and convincing.

F. Suggested improvements: experiments, data for possible revision

None

G. References: appropriate credit to previous work?

Yes. One small comment: the study by Lalueza Fox et al. (2011, PNAS) is cited at the start of the article (in relation to community size), but I think it could be worth referring to this again in the discussion or conclusion, specifically at line 374-376 and around 387, since this study also provides evidence for female biased migration, and is relevant to understanding whether such social organization patterns were local or widespread.

<https://www.pnas.org/doi/full/>

See response to reviewer 3.

H. Clarity and context: lucidity of abstract/summary, appropriateness of abstract, introduction and conclusions

The article as a whole is clear. I have a few specific suggestions with regard to clarity at the sentence level, listed below.

58-60 (abstract) 'These genetic data represent the earliest known example of social organization in a Neanderthal population at the easternmost extent of their known range.' This should perhaps be rephrased because it can be read to suggest social organization was absent before, while what is meant is that this is the first information available about the way they organized themselves socially.

See response to reviewer 3.

110-112 This sentence is unclear. It might be helpful to add 'however' to indicate that this sentence outlines a different pattern from the previous one, and also to mention which individual from Denisova Cave is being discussed (it could be read as still discussing Denisova 5).

We have changed the sentence to make it clear that we are now discussing another individual:

A first generation offspring (Denisova 11) of a Neanderthal mother and a Denisovan father revealed that the Neanderthal mother is more closely related to Chagyrskaya 8 than she is to other Neanderthals

381-383 These sentences may be confusing and might benefit from rephrasing; it reads as if the coalescence time for the y-chromosome was part of evidence for small communities, whereas from the preceding paragraphs I understood this to be primarily relevant for migration patterns.

We agree that this wording was a bit unclear. We clarified this paragraph to

We present genetic data from 13 Neanderthals, making this the largest genetic study to date of a Neanderthal population. For the first time, we document familial relationships between Neanderthals, including a father-and-daughter pair.

The high degree of homozygosity in all individuals is similar to what is seen in mountain gorillas³⁹ consistent with Neanderthals in the Altai living in small communities. In addition, based on the shorter average coalescent time for the Y-chromosomes than for the mtDNA and shared mtDNA variants between Chagyrskaya and Okladnikov individuals, we suggest that these small Neanderthal communities were predominantly linked by female migration

There are also two sentences in the supplementary materials which are not clear: Supplementary materials p. 5: 'We also identify two remains that show a second degree relationship - one in the subseries IIa and another in subseries IIa. These remains are

Chagyrskaya 1 found in the colluvial sediments and Chagyrskaya 60 found in the intact sediments (SI Figure 1.2 and SI Figure 1.3).’ This should read ‘subseries IIa and another in subseries IIb’ I assume.

We thank the reviewer for spotting this error. Chagyrskaya 1 is found in subseries IIb (colluvial sediments) and Chagyrskaya 60 is found in the intact sediment layer subseries IIa. We have corrected this in the supplementary material.

SM p. 6 ‘Their population size was probably small, based on genetic data ((Mafessoni et al., 2020); Supplementary Sections 8 and 9) and the limited area of living space in the cave, so it may have been used mainly as a camp for the selective hunting and consumption of bison and horses, with a preference for non-adult individuals and females (Kolobova et al., 2019).’ What does population size have to do with area of the living space or use as hunting camp?

We have now clarified the section to read:

Kolobova et al., 2019 have suggested that given the limited area of living space in the cave, it may have been used mainly as a camp for the selective hunting and consumption of bison and horses, with a preference for non-adult individuals and females. The small number of individuals is consistent with genetic analyses indicating that the Neandertal community in the region was likely small (Mafessoni et al., 2020; Supplementary Sections 8 and 9).

Reviewer Reports on the First Revision:

Referees' comments:

Referee #1 (Remarks to the Author):

Thank you for your attention to detail and careful consideration of my comments. I am very happy with the scientific content of the revised manuscript - but a final proofread would be beneficial.

Referee #2 (Remarks to the Author):

The authors have suitably addressed my comments.

Referee #3 (Remarks to the Author):

I thank the authors for their response. Most of my comments have been fully addressed, however, there are several outstanding issues that I feel were not properly dealt with. I elaborate on these in more detail below, in case the meaning of my original requests were misunderstood by the authors due to their brevity or poor phrasing.

1. Inbreeding (Altai versus Chagyrskaya).

The authors did not engage with my comment here. I asked them to be explicit about the results from Prüfer et al. (2014) and how these interact with their own results. Prüfer et al. did not simply conclude her parents were "closely related", which is an inexact term (see note below), they concluded they were second degree relatives based on inbreeding simulations:

"We conclude that the parents of this Neanderthal individual were either half-siblings who had a mother in common, double first cousins, an uncle and a niece, an aunt and a nephew, a grandfather and a granddaughter, or a grandmother and a grandson."

Please be explicit about this result as it is an important benchmark for the Chagyrskaya population. Furthermore, be explicit about how this result impacts your own and visa versa. Are you suggesting that 2nd degree matings were a reoccurring feature of Eastern Neanderthal populations over tens of thousands of years? If true, this is quite remarkable from both an anthropological and biological perspective and needs to be emphasized.

Alternatively, do the authors think that the Chagyrskaya ROH profiles are the result of more complex pedigrees with multiple inbreeding loops, which Prüfer et al. did not consider in their simulations? I note that one study on Mountain Gorillas estimates that about 90% of the parents of offspring are not related on the level of half-sibling or higher (Vigilant et al. 2015), although the results from Xue et al. note that their ROH profiles are more extreme than the Altai Neanderthal.

Why did the authors not carry out similar inbreeding simulations to see what scenarios best fit the ROH length distributions at Chagyrskaya? Comparing the results in SI Table 9.2 to the simulations from Prüfer et al. (Figure S10.5 in that paper), it seems the number and length of tracts above 10 cM for most Chagyrskaya individuals fits comfortably into the range seen for simulated 2nd degree matings in Prüfer et al. However, I think the Skov and colleagues use a newer method to detect ROH segments (admixfrog)? The Chagyrskaya 8 paper (Mafessoni et al. 2020) seems to follow the method from Prüfer et al. and finds that:

“About 12.9% of the Chagyrskaya 8 genome is covered by HBD tracts of size 2.5cM to 10cM. Such HBD tracts cover 5.7% and 6.2% of the Denisova 5 and Vindija 33.19 Neandertal genomes, respectively, and 2.6% of the Denisova 3 genome. Longer HBD tracts (>10cM) account for 6.4% of the Chagyrskaya 8 genome, 10.5% of the Denisova 5 genome”

The current study finds that 9.2% of the Chagyrskaya 8 (Chagyrskaya F) genome is in runs over 10cM and 9.5% in runs between 2.5-10 cM. The total ROH >2.5cM is similar for both studies (18.7-19.3%), but newer method seems to be combining more short runs into longer runs.

It is vital for the authors to engage with this past body of work, especially where their results or conclusions conflict with prior findings. Abundant inbreeding in a single community is headline aspect of this paper and needs to be explored fully.

Finally, if 2nd degree inbreeding is indeed the norm in these populations, please be clear that this has been accounted for in the detection of 1st and 2nd degree relatives by READ.

NOTE: The authors need to be clearer on what is meant by “closely related”, especially for the non-specialist reader. For example, the abstract states that “All Chagyrskaya individuals are very closely related”, while the parents of Denisova 5 are only “closely related”. Chagyrskaya D is also “closely related” to his daughter, a possible 4th degree maternal relative, and potentially himself (Chagyrskaya E).

2. Multi-generational female line transmissions

I am confused as to why the authors are not willing to provide some interpretation of the Chagyrskaya D and C result, given that a major focus of the paper is female mobility and a finite set of scenarios exist that explain this relationship. Perhaps I didn't phrase my original comment very well. To be clear, by multiple transmissions along the female line I meant scenarios that include consecutive females across at least two generations (e.g. Female->Female->Offspring)

I don't think it is a “strong conclusion” to suggest that if two individuals are close maternal relatives beyond the second degree there was at least one incidence of multiple transmissions along the female line within this community. The only alternative to this I can conceive is the return of the female descendants of females who migrated (being mindful that heteroplasmies typically persist for less than three generations) or the return of Chagyrskaya D and/or C to their mothers' ancestral

community (less likely given the modelling results).

Maybe I'm missing something here? If so, perhaps the authors could explain to both myself and the reader why their data do not support these conclusions. As the authors say themselves, such scenarios are consistent with their modelling. Perhaps they are not fully confident in the heteroplasmy analysis and inferred maternal line relationship between Chagyrskaya D and C?

I understand the need for caution, but these are fascinating results and deserve to be robustly synthesized and interpreted for a broad readership, even if no definite conclusions are drawn.

3. Providing context to the non-specialist reader

The authors provided useful context and justification for me in their response letter, however, some of this would also be of benefit to the paper's readership. I wonder could they add some of this text to the paper itself. Main points below:

- I was not aware of the significance of sampling both deciduous and permanent teeth from the same individual. As this paper is intended for a wide audience, could the authors be more clear as to the useful lesson learned here.

- The same goes for the use of heteroplasmies. This is indeed an underutilized approach in the field, but this is not obvious to the reader in the main text. Could the authors please highlight the innovativeness of their approach in lines 179-185.

- That most great ape data they use comes from apes in captivity.

- Please be explicit that you are suggesting that the Chagyrskaya Neanderthals' "social organization is similar to the primates with similar ratios", as you have stated in your response letter. Following from this, I again ask for the authors to provide some context on the social organisation of those primates. Why do different great ape populations show such differences in their ratios of mitochondrial DNA to Y-chromosome diversity (Extended Data Figure 6: e.g. Bornean versus Sumatran Orangutans, Eastern versus Western Chimpanzees)? If the authors do not know the answer to this, then perhaps they should be wary of using them as comparative data. In the supplementary, the authors note that the three populations most similar to Chagyrskaya in terms of Y-to-MT diversity ratios are highly fragmented and endangered/critically endangered, but this seems more relevant to absolute estimates of diversity. Bornean Orangutans are also critically endangered but show little difference in their mtDNA and Y chromosome coalescence times.

To summarize, if the authors are indeed using empirical data from the apes to support their conclusion of female biased dispersal from simulations, then they need to make this case clearly. If the ape data cannot be used as support (e.g. confounding effects of captive-breeding programs), then they should be clear on this.

Minor Issues

Lines 210-220: It is confusing to discuss the relationship between D and C simultaneously with that between D and E, as the authors draw quite different conclusions (possible 4th degree relative versus 1st degree or identical). To the reader, the first lines of this paragraph seem to imply that Chagyrskaya E is a separate individual and close maternal relative to D (~4th degree). It would be better to discuss these relationships one at a time to avoid any misreadings.

Line 223: please give the molecular sex of Chagyrskaya A and L here as it is of interest to reader.

Typo: industri->industry

Figure1: Chagyrskaya 8 is not coloured as Chagyrskaya. Chagyrskaya F is also included, but I thought this individual was a combination of Chagyrskaya 8 and 12. However, Chagyrskaya F appears to have lower coverage than Chagyrskaya 8.

REFS

Mafessoni, F. et al. A high-coverage Neandertal genome from Chagyrskaya Cave. *Proc. Natl. Acad. Sci. U. S. A.* 117, 15132–15136 (2020).

Prüfer, K. et al. The complete genome sequence of a Neanderthal from the Altai Mountains. *Nature* 505, 43–49 (2014).

Vigilant et al. Reproductive competition and inbreeding avoidance in a primate species with habitual female dispersal. *Behavioral Ecology and Sociobiology* 69, 1163–1172 (2015).

Xue, Y. et al. Mountain gorilla genomes reveal the impact of long-term population decline and inbreeding. *Science* 348, 242–245 (2015).

Referee #4 (Remarks to the Author):

The authors have successfully and comprehensively addressed the comments of the reviewers. Among other things, the changes bring out the key implications of the research more clearly in the abstract and conclusion.

I note that a type has been introduced in the corrections - in several places 'industri' is written, this should be 'industry'.

Author Rebuttals to First Revision:

We would once again like to thank the reviewers for their thoughtful comments and efforts towards improving our manuscript. We address comments specific to each reviewer below and use **red** to highlight the reviewer's question, **blue** to highlight our answer and **green** to highlight changes to the manuscript/supplementary information.

In regards to the main text we have kept track changes in, so sentences which have been added/changed are highlighted in **green** and sentences which have been moved to the supplement/deleted are highlighted in **red**.

Referee #1 (Remarks to the Author):

Thank you for your attention to detail and careful consideration of my comments. I am very happy with the scientific content of the revised manuscript - but a final proofread would be beneficial.

Referee #2 (Remarks to the Author):

The authors have suitably addressed my comments.

Referee #4 (Remarks to the Author):

The authors have successfully and comprehensively addressed the comments of the reviewers. Among other things, the changes bring out the key implications of the research more clearly in the abstract and conclusion.

I note that a type has been introduced in the corrections - in several places 'industri' is written, this should be 'industry'.

We have now changed the misspelling of industri to industry throughout.

Referee #3 (Remarks to the Author):

I thank the authors for their response. Most of my comments have been fully addressed, however, there are several outstanding issues that I feel were not properly dealt with. I elaborate on these in more detail below, in case the meaning of my original requests were misunderstood by the authors due to their brevity or poor phrasing.

1. Inbreeding (Altai versus Chagyrskaya).

The authors did not engage with my comment here. I asked them to be explicit about the results from Prüfer et al. (2014) and how these interact with their own results. Prüfer et al. did not simply conclude her parents were “closely related”, which is an inexact term (see note below), they concluded they were second degree relatives based on inbreeding simulations:

“We conclude that the parents of this Neanderthal individual were either half-siblings who had a mother in common, double first cousins, an uncle and a niece, an aunt and a nephew, a grandfather and a granddaughter, or a grandmother and a grandson.”

Please be explicit about this result as it is an important benchmark for the Chagyrskaya population. Furthermore, be explicit about how this result impacts your own and visa versa. Are you suggesting that 2nd degree matings were a reoccurring feature of Eastern Neanderthal populations over tens of thousands of years? If true, this is quite remarkable from both an anthropological and biological perspective and needs to be emphasized.

Alternatively, do the authors think that the Chagyrskaya ROH profiles are the result of more complex pedigrees with multiple inbreeding loops, which Prüfer et al. did not consider in their simulations?

We thank the referee for clarifying their previous comment, and apologize for the misunderstanding. Overall, our stance is that the levels of ROH observed in the Chagyrskaya and Denisova Neanderthals can be explained by small community sizes, as was already suggested by Mafessoni et al (2020) (Their Figure 3B and Supplement 8).

It is very hard to distinguish the Mafessoni et al. scenarios and the Prufer et al. scenarios based on a single individual. However, they make very different predictions regarding multiple individuals, which is why we do not need to rely on simulations for this analysis:

- The Prufer et al. scenario of an offspring of 2nd-degree relatives against a background of less ROH (effective population size of 700 individuals) assumes that the studied individual is in some sense unusual, and that for most individuals in the population, the ROH would look like the background (i.e. much lower levels of ROH)
- In contrast, the small-community model of Mafessoni et al. predicts that everyone in the population has similar, large levels of ROH. This is because under such scenarios, the pedigrees will indeed have many inbreeding loops due to the very limited population sizes. As we find that all individuals have ROH, we think that a small community size is the better explanation.

We did not mean to suggest that 2nd-degree matings were recurrent throughout Neanderthal history, and we appreciate that the previous wording could have been interpreted as that. We therefore opted to rewrite and extend this paragraph to explicitly address the scenarios put forward by Prufer et al. and Mafessoni et al:

We investigated the community and population size of the Chagyrskaya Neanderthals through time using genomic segments of homozygosity from eight individuals (those with >0.9 fold genomic coverage, SI 9). Long segments of homozygosity (>10cM) in an individual imply that their parents shared a very recent common ancestor around ten generations ago, and hence were likely part of a small community^{5,37}. In addition, the overall proportion of the genome in intermediate length segments of homozygosity (2.5-10 cM) is informative about the size of the population over a slightly longer time-frame (~10-40 generations).

Previous analyses of high-coverage Neanderthal genomes from the Altai mountains revealed that around 16.7% of the genome of Denisova 5², and 19.3% of the genome of Chagyrskaya 8⁵ were in intermediate and long segments of homozygosity. One explanation for these patterns is that their parents were 2nd-degree relatives² against a background of unrelated individuals, in which case we would expect most other individuals to have fewer homozygous segments. Alternatively, these data could be due to small local communities⁵, in which case all individuals except recent immigrants and their descendants would have extensive segments of homozygosity.

In all eight individuals with sufficient coverage, we observed 1.6-14.9% of the genome in long segments and 9.5-20.5% in intermediate segments of homozygosity, respectively. (Figure 2a, SI Table 9.2). We note that both proportions were likely under-estimates due to difficulties in identifying runs of homozygosity at lower coverages (SI Table 9.1). Since we find high amounts of homozygosity in all individuals, we conclude that the local community size of the Chagyrskaya Neanderthals was small. The amount of homozygosity is also similar to the amount found in the genomes of present-day mountain gorillas (Figure 2b)³⁸, an endangered species that lives in small communities of 4-20 individuals³⁹, where it has been observed that matings between 2nd-degree related individuals are rare⁴⁰.

I note that one study on Mountain Gorillas estimates that about 90% of the parents of offspring are not related on the level of half-sibling or higher (Vigilant et al. 2015), although the results from Xue et al. note that their ROH profiles are more extreme than the Altai Neanderthal.

We thank the reviewer for this comment. As clarified above, we too favor a scenario where high levels of ROH are achieved without matings of 2nd-degree relatives, so we think the Neanderthal data is consistent with the observations in mountain gorillas, and we adjusted the main text to include a citation to the Vigilant paper (as citation 40):

The amount of homozygosity is also similar to the amount found in the genomes of present-day mountain gorillas (Figure 2b)³⁸, an endangered species that lives in small communities of 4-20 individuals³⁹, where it has been observed that matings between 2nd-degree related individuals are rare⁴⁰.

Why did the authors not carry out similar inbreeding simulations to see what scenarios best fit the ROH length distributions at Chagyrskaya?

We note that Mafessoni et al. (2020) showed that these large amounts of long ROH are also compatible with a small community model (see in particular their Figure S8.14), so they cannot easily be used to distinguish small population size from the offspring of a second degree mating, which is why we did not carry out simulations to this regard. In contrast, we think that the differing predictions regarding the number of individuals showing ROH between the two models are a much simpler way to distinguish them.

Comparing the results in SI Table 9.2 to the simulations from Prüfer et al. (Figure S10.5 in that paper), it seems the number and length of tracts above 10 cM for most Chagyrskaya individuals fits comfortably into the range seen for simulated 2nd degree matings in Prüfer et al. However, I think the Skov and colleagues use a newer method to detect ROH segments (admixfrog)? The Chagyrskaya 8 paper (Mafessoni et al. 2020) seems to follow the method from Prüfer et al. and finds that:

“About **12.9% of the Chagyrskaya 8** genome is covered by HBD tracts of size 2.5cM to 10cM. Such HBD tracts cover 5.7% and 6.2% of the Denisova 5 and Vindija 33.19 Neanderthal genomes, respectively, and 2.6% of the Denisova 3 genome. Longer HBD tracts (>10cM) account for **6.4% of the Chagyrskaya 8** genome, 10.5% of the Denisova 5 genome”

The current study finds that **9.2% of the Chagyskaya 8** (Chagyskaya F) genome is in runs over 10cM and **9.5% in runs between 2.5-10 cM**. The total ROH >2.5cM is similar for both studies (18.7-19.3%), but newer method seems to be combining more short runs into longer runs.

The reason we use different methodology is because the data quality is very different between the studies. We have information for 643,472 capture sites and most individuals are low coverage. This is in contrast to the shotgun-sequenced high-coverage genomes analyzed by Prufer et al and Mafessoni et al, which allow for much preciser calling of ROH boundaries (for context, see also the detailed analysis on this problem by Ringbauer et al. 2021 <https://www.nature.com/articles/s41467-021-25289-w.pdf>, who analyze better data with a method that relies on a large reference panel, which we do not have for Neanderthals).

To further make the limitations of our data and approach transparent, we added a summary table **SI Table 9.1** but the reviewer is correct that our method combines short runs into longer runs. We now provide a new supplemental Figure **SI Figure 9.1** - which allows the reader to visually inspect the differences between the two approaches.

It is vital for the authors to engage with this past body of work, especially where their results or conclusions conflict with prior findings. Abundant inbreeding in a single community is headline aspect of this paper and needs to be explored fully.

Finally, if 2nd degree inbreeding is indeed the norm in these populations, please be clear that this has been accounted for in the detection of 1st and 2nd degree relatives by READ.

It is true that ROH will bias relatedness estimates. To account for this, we developed a new method for relatedness estimation (KIN), that takes ROH into account explicitly. However, as this turned out to be a major endeavor that required extensive testing of the resulting method, we decided to publish this method independently. Using KIN, we show that we can recapture the findings described here while taking the ROH into account explicitly. We attach the submitted manuscript for KIN for reference.

NOTE: The authors need to be clearer on what is meant by “closely related”, especially for the non-specialist reader. For example, the abstract states that “All Chagyskaya individuals are very closely related”, while the parents of Denisova 5 are only “closely related”. Chagyskaya D is also “closely related” to his daughter, a possible 4th degree maternal relative, and potentially himself (Chagyskaya E).

We appreciate this comment. We no longer use the term “very closely related”, and only use “closely related” to describe pairs of individuals that are more similar to each other than would be expected from two random individuals from the same community ($p_0 < 1$) or when describing individuals who share a heteroplasmy.

2. Multi-generational female line transmissions

I am confused as to why the authors are not willing to provide some interpretation of the Chagyrskaya D and C result, given that a major focus of the paper is female mobility and a finite set of scenarios exist that explain this relationship. Perhaps I didn't phrase my original comment very well. To be clear, by multiple transmissions along the female line I meant scenarios that include consecutive females across at least two generations (e.g. Female->Female->Offspring)

I don't think it is a “strong conclusion” to suggest that if two individuals are close maternal relatives beyond the second degree there was at least one incidence of multiple transmissions along the female line within this community. The only alternative to this I can conceive is the return of the female descendants of females who migrated (being mindful that heteroplasmies typically persist for less than three generations) or the return of Chagyrskaya D and/or C to their mothers' ancestral community (less likely given the modelling results).

Maybe I'm missing something here? If so, perhaps the authors could explain to both myself and the reader why their data do not support these conclusions. As the authors say themselves, such scenarios are consistent with their modelling. Perhaps they are not fully confident in the heteroplasmy analysis and inferred maternal line relationship between Chagyrskaya D and C?

I understand the need for caution, but these are fascinating results and deserve to be robustly synthesized and interpreted for a broad readership, even if no definite conclusions are drawn.

We misunderstood the question - our apologies. We thought the reviewer initially asked if our data allows us to differentiate between the following two scenarios 1) The shared heteroplasmy is due to multi generation female line transmission and 2) the return of a female descendants of a female who migrated. Our observation is consistent with both (and the most parsimonious is probably multi generation female line transmission as the reviewer points out). We have now added this to the discussion on female migration:

This result suggests that female-biased migration was a major factor in the social organization of the Chagyrskaya Neanderthal community. However the shared heteroplasmy between *Chagyrskaya C* and *Chagyrskaya D* suggests that at least some females remained with the group they were born in.

3. Providing context to the non-specialist reader

The authors provided useful context and justification for me in their response letter, however, some of this would also be of benefit to the paper's readership. I wonder could they add some of this text to the paper itself. Main points below:

- I was not aware of the significance of sampling both deciduous and permanent teeth from the same individual. As this paper is intended for a wide audience, could the authors be more clear as to the useful lesson learned here.

We have now changed the main

We found a deciduous tooth (*Chagyrskaya 19*) and two permanent teeth (*Chagyrskaya 13* and *Chagyrskaya 63*). Surprisingly, despite their different development stages, the genetic data suggests that they belonged to the same individual (*Chagyrskaya G*, average $p_0=0.53$, Extended Data Figure 3A).

- The same goes for the use of heteroplasmies. This is indeed an underutilized approach in the field, but this is not obvious to the reader in the main text. Could the authors please highlight the innovativeness of their approach in lines 179-185.

We agree that this approach is underutilized. However, given that it is not novel (see refs), we cannot really claim its innovation. Instead, we hope the utility it provided us here will promote further adoption.

- That most great ape data they use comes from apes in captivity.

- Please be explicit that you are suggesting that the Chagyrskaya Neanderthals' "social organization is similar to the primates with similar ratios", as you have stated in your response letter.

Following from this, I again ask for the authors to provide some context on the social organisation of those primates. Why do different great ape populations show such differences in their ratios of mitochondrial DNA to Y-chromosome diversity (Extended Data Figure 6: e.g. Bornean versus Sumatran Orangutans, Eastern versus Western Chimpanzees)?

If the authors do not know the answer to this, then perhaps they should be wary of using them as comparative data. In the supplementary, the authors note that the three populations most similar to Chagyrskaya in terms of Y-to-MT diversity ratios are highly fragmented and endangered/critically endangered, but this seems more relevant to absolute estimates of diversity. Bornean Orangutans are also critically endangered but show little

difference in their mtDNA and Y chromosome coalescence times.

To summarize, if the authors are indeed using empirical data from the apes to support their conclusion of female biased dispersal from simulations, then they need to make this case clearly. If the ape data cannot be used as support (e.g. confounding effects of captive-breeding programs), then they should be clear on this.

This is a good point and we agree that the comparisons with the apes are limited in their scope. However, given that the alternative is to just rely on our *in silico* modeling, we maintain that the apes data provide a useful sanity check that such extreme Y-to-MT diversity ratios exist in other primates.

We thus modified the main text and supplement to make the limitations of the great-ape comparisons more explicit: First, we address the issue that there could be actual differences in behavior between different subspecies of great apes. We have added the following overview of social organization to the supplementary section:

Apes too show substantial variation in their social organization. Gorillas, chimpanzees and bonobos all tend to show male philopatry and female dispersal, but exceptions have been observed (Douadi et al. 2007)(Schubert et al. 2011; Lukas et al. 2005). Gorillas commonly have one-male groups (Harcourt

and Stewart 2007) with female dispersal (Bradley et al. 2007; Parnell 2002; Robbins et al. 2009) although, male dispersal can also occur

Chimpanzee and bonobo communities also commonly show male philopatric tendencies while females migrate (Langergraber et al. 2007; Nishida et al. 2003; McCarthy et al. 2015) (Hohmann 2001; Eriksson et al. 2006) although, this does not seem to be the case in some Western chimpanzee communities where there is both female and male migration (Schubert et al. 2011; Lukas et al. 2005).

In orangutans females philopatric tendencies and male dispersal have been observed in both Sumatran (Singleton and van Schaik 2002; Nater et al. 2011) and Bornean orangutans (van Noordwijk et al. 2012; Nietlisbach et al. 2012; Nater et al. 2011). However, at least one study also found evidence for both male and female philopatry in the latter.

Second the differences could be due to low sample sizes and non-random sampling of individuals. Below we show the number of mitochondrial sequences and Y chromosome sequences for the great apes (has been added as **SI Table 8.1**). Note the number of Y chromosomes will be equal to the number of males and the number of mitochondrial sequences will be equal to the number of males and females. We have highlighted places where there are either only two Y chromosomes (meaning we can only make one Y chromosome comparison) or places where less than 50% of individuals are wildborn - which could also introduce bias. We have removed these species in **Extended Figure 6** but keep them in **SI Figure 8.3** for completeness.

Common name	Species name	mt		Y chromosomes	
		n	Wildborn	n	Wildborn
Eastern lowland gorilla	Gorilla beringei graueri	7	89.0%	2	100.0%
Western lowland gorilla	Gorilla gorilla gorilla	43	71.0%	7	85.7%
Mountain gorilla	Gorilla beringei beringei	8	100.0%	3	100.0%
Bonobo	Pan paniscus	16	75.0%	4	100.0%
Nigerian chimpanzee	Pan troglodytes ellioti	6	100.0%	4	100.0%
Eastern chimpanzee	Pan troglodytes schweinfurthii	7	100.0%	3	100.0%
Central chimpanzee	Pan troglodytes troglodytes	6	100.0%	2	100.0%
Western chimpanzee	Pan troglodytes verus	8	38.0%	8	28.6%
Sumatran orangutan	Pongo abelii	6	50.0%	4	25.0%

Bornean orangutan	Pongo pygmaeus	8	33.0%	2	50.0%
-------------------	----------------	---	-------	---	-------

While these limitations of the data make it hard to gauge the certainty in individual data points, the finding that the observed social organization largely matches the observed Y-to-mt ratios (**Extended Figure 6**) does lead to some credence and utility of these comparisons. We address these limitations by adding the following paragraph to the main text.

We caution that similar ratios between apes and Neanderthals do not necessarily mean that the communities have the same social organization, as there are multiple caveats. First, the great ape data is very heterogeneous: For example, some great apes were wildborn, others were born in captivity (i.e. in artificial communities) and often the sample sizes are very small (**SI Table 8.1**). Second, several different scenarios may lead to similar Y-to-mt ratios.

Minor Issues

Lines 210-220: It is confusing to discuss the relationship between D and C simultaneously with that between D and E, as the authors draw quite different conclusions (possible 4th degree relative versus 1st degree or identical). To the reader, the first lines of this paragraph seem to imply that Chagyrskaya E is a separate individual and close maternal relative to D (~4th degree). It would be better to discuss these relationships one at a time to avoid any misreadings.

Line 223: please give the molecular sex of Chagyrskaya A and L here as it is of interest to reader.

Chagyrskaya A is a male and Chagyrskaya L is female - we have now added this to the main text.

Typo: industri->industry

Has been changed

Figure1: Chagyrskaya 8 is not coloured as Chagyrskaya. Chagyrskaya F is also included, but I thought this individual was a combination of Chagyrskaya 8 and 12. However, Chagyrskaya F appears to have lower coverage than Chagyrskaya 8.

We apologize for the confusion. The color blue is Chagyrskayas from this study (meaning this does not include Chagyrskaya 8). We now state this explicitly in the figure legend. It is also correct that Chagyrskaya 12 and Chagyrskaya 8 are remains from the same individual which is called Chagyrskaya F. We do not combine the genomic data here because we want to illustrate how many whole genomes this study adds to the literature.

Reviewer Reports on the Second Revision:

Referees' comments:

Referee #3 (Remarks to the Author):

I thank the authors for their very thorough and thoughtful response to my second round of review. I'm satisfied that my concerns have been addressed and have no further comments except to say congratulations on what I'm sure will be a highly impactful paper.